# Cultural hitchhiking and competition between patrilineal kin groups explain the post-Neolithic Y-chromosome bottleneck

Tian Chen Zeng[1,2], Alan J. Aw[2,3] & Marcus W. Feldman[3]

In human populations, changes in genetic variation are driven not only by genetic processes, but can also arise from cultural or social changes. An abrupt population bottleneck specific to human males has been inferred across several Old World (Africa, Europe, Asia) populations 5000–7000 BP. Here, bringing together anthropological theory, recent population genomic studies and mathematical models, we propose a sociocultural hypothesis, involving the formation of patrilineal kin groups and intergroup competition among these groups. Our analysis shows that this sociocultural hypothesis can explain the inference of a population bottleneck. We also show that our hypothesis is consistent with current findings from the archaeogenetics of Old World Eurasia, and is important for conceptions of cultural and social evolution in prehistory.

[1] Department of Sociology, Stanford University, Stanford, CA 94305, USA. [2] Mathematical and Computational Science Program, Stanford University, Stanford, CA 94305, USA. [3] Department of Biology, Stanford University, Stanford, CA, USA. These authors contributed equally: Tian Chen Zeng, Alan J. Aw. Correspondence and requests for materials should be addressed to M.W.F. (email: mfeldman@stanford.edu)

n human populations, changes in genetic variation are driven not only by genetic processes, but can also arise from cultural or social changes[1,2]. Cultural factors and processes can influence migration patterns and genetic isolation of populations, and can be responsible for the patterns of genetic variation as a result of gene-culture co-inheritance (e.g. a preference of cousin marriage[3]). Understanding how social and cultural processes affect the genetic patterns of human populations over time has brought together anthropologists, geneticists and evolutionary biologists, and the availability of genomic data and powerful statistical methods widens the scope of questions that analyses of genetic information can answer.

Using a data set of 125 Y-chromosome sequences from modern humans, Karmin et al.[4] inferred an intense bottleneck in Y-chromosomes in various geographical regions of the Old World around 5000–7000 BP (Fig. 1), suggesting a decline in the male effective population size during the Neolithic to approximately one-twentieth of its original level before the Neolithic in regions

including Africa, Europe, Asia and the Middle East. In the same study, mitochondrial sequences indicated a continual increase in population size from the Neolithic to the present, suggesting extreme divergences between the demographic size of male and female populations in the bottleneck period. Another study of Y-chromosomes of modern humans across Europe confirmed the population bottleneck[5]. Poznik et al.[6] constructed a phylogenetic tree and observed star-like expansions of Y-chromosome lineages in the proposed bottleneck period; compare the clades indicated in Fig. 2 with the tree shape in Fig. 3, particularly under the O, R1a and R1b haplogroups.

The inferred bottleneck and associated star-like expansions on the phylogenetic tree were confined to male-inherited Y-chromosomes (Fig. 3) and were not apparent in female-inherited mitochondrial DNA. This suggests that around 5000–7000 BP, coinciding with the post-Neolithic period in each region of the Old World for which the bottleneck was found, there were minor changes in the number of reproducing females and a more stable

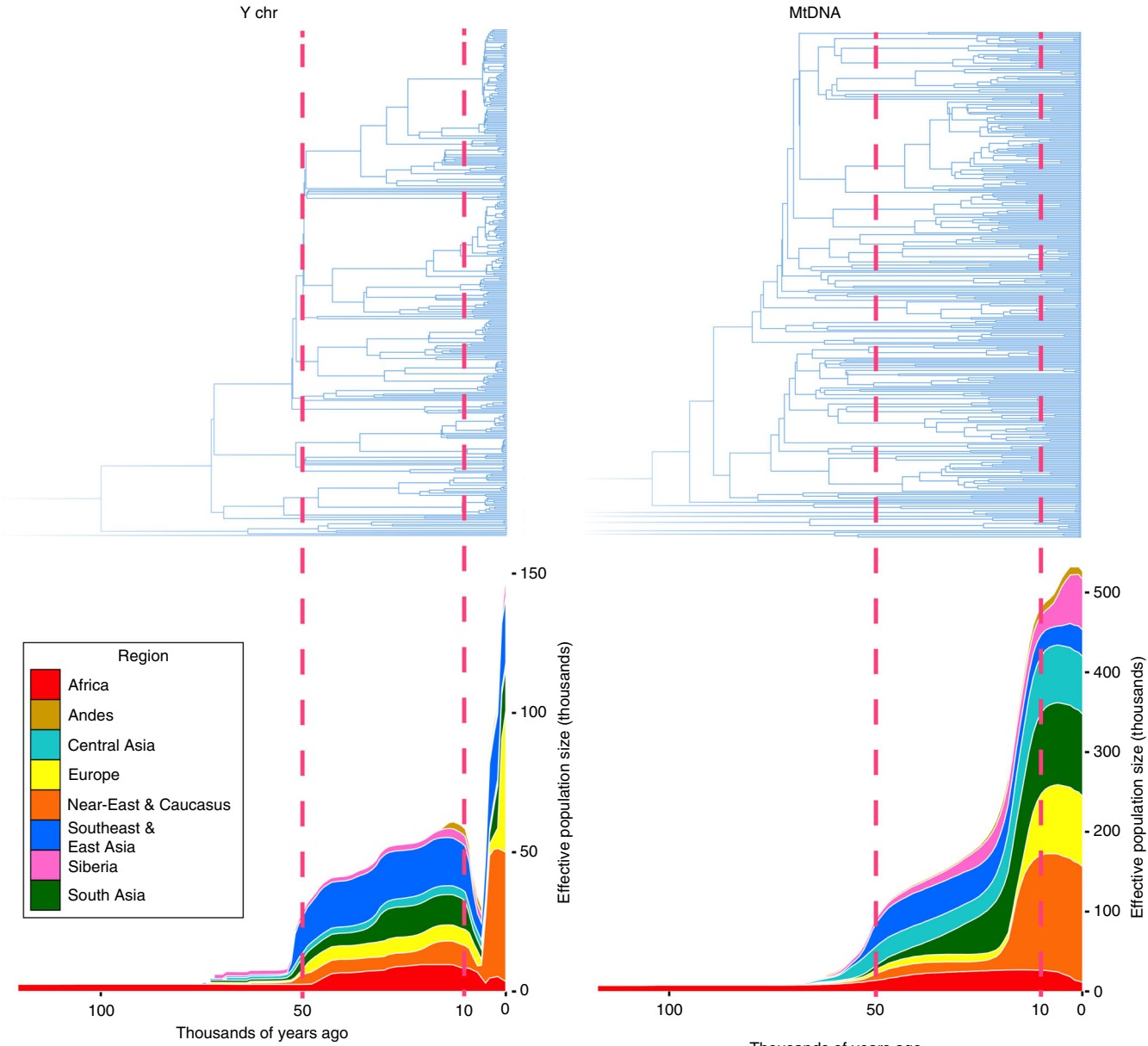

**Fig. 1** Cumulative Bayesian skyline plots of Y-chromosome and mtDNA diversity by world regions. Reprinted from Karmin et al.[4] with permission from Monika Karmin and under a Creative Commons License (Attribution-NonCommercial 4.0 International)

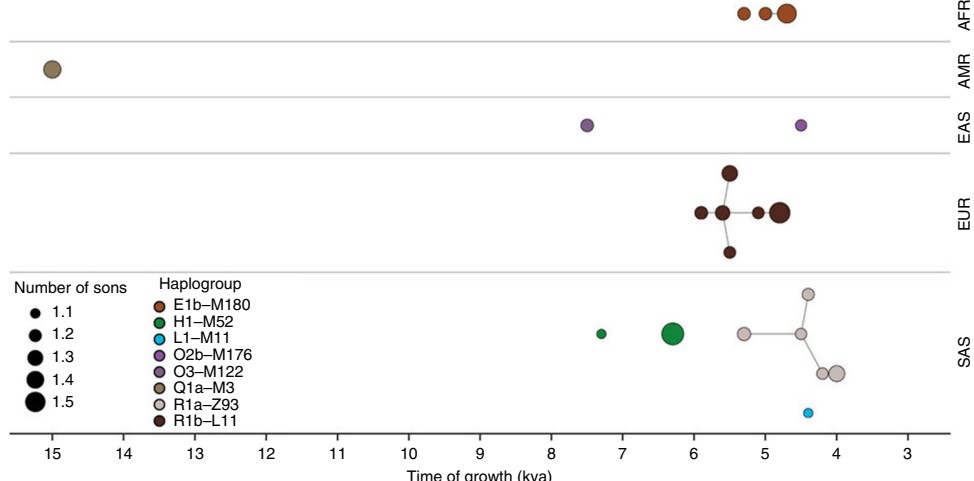

**Fig. 2** Explosive male-lineage expansions of the last 15,000 years. Each circle represents a phylogenetic node whose branching pattern suggests rapid expansion. The horizontal axis indicates the timings of the expansions, and circle radii reflect growth rates, namely the minimum number of sons per generation, as estimated by a two-phase growth model of Poznik et al.[6]. Nodes are grouped by continental superpopulation (AFR: African, AMR: admixed American, EAS: East Asian, EUR: European, SAS: South Asian) and coloured by haplogroup. Line segments connect phylogenetically nested lineages. Reprinted from Poznik et al.[6] with permission from Carlos D. Bustamante and Nature Publishing Group

female population, whereas dramatic reduction in the number of reproducing males occurred. At the same time, pronounced differences in tree shape appeared, as seen in Fig. 1.

We propose a hypothesis that explains the apparent mismatch in population dynamics of males and females, and hence the male bottleneck. Before explaining our proposal, we examine three hypotheses that we believe cannot explain the male bottleneck that has been inferred from Y-chromosomal data.

First, it is possible that ecological or climatic factors cause sex-specific demographic change. Some ecological or climatic factors such as stress or disease may impact male infant mortality and sex ratio at birth[7,8]. However, such effects are small and cannot account for the 1:17 disparity[4] between male and female population sizes inferred from the uniparental data.

Second, Neolithic founder effects from small populations of male Neolithic pioneers could create the appearance of a bottleneck in modern Y-chromosomes. Agriculture-driven population expansions from centres of domestication have been hypothesized[9,10], and have been confirmed through archaeogenetics[11,12]. Linguists and anthropologists have theorized that male-biased population expansions suggested by genetic data may account for the distribution of language families and cultures in the 'Father Tongues Hypothesis'[13–15]. Population expansions from small founding groups of Neolithic males would carry both male uniparental markers and cultural packages that included Neolithic subsistence strategies, displacing hunter-gatherer Y-chromosomes and cultural practices, while mitochondrial chromosomes would remain diverse as large local populations of hunter-gatherer females became assimilated into the expanding agricultural population in regional admixture events. This would create a selection bias among modern uniparental markers that produces the appearance of a bottleneck specific to the Y-chromosome.

However, archaeogenetic evidence contradicts this hypothesis. In West Eurasia, the precursors of the first Neolithic farming populations already had larger population sizes[16,17] than hunter gatherers from areas that received agriculturalist migration, validating archaeological and anthropological theories that emphasize the role of population pressure among sedentary hunter gatherers in driving the Neolithic transition in centres of domestication[18–21]. Neolithic precursor populations had greater

diversity of both Y and mitochondrial chromosomes than hunter-gatherer populations, which tended to have very homogeneous uniparental markers[22,23]. In fact, most mitochondrial lineages in modern Europeans derive from first farmers, and have a Neolithic time-depth[24].

Therefore, this hypothesis cannot be accurate, at least for West Eurasia, as the population of Neolithic founding males was comparatively large, and greater diversity in mitochondrial lineages in modern populations was not produced by retention of local hunter-gatherer maternal lineages. Genomes from ancient hunter gatherers in East Asia[25], Neolithic farmers in Iran[16,17], and ancient African hunter gatherers[26,27], allow us to infer, through ancient–modern comparisons, that the genetic contribution from Neolithic populations to present-day farming or pastoralist Africans and South and East Asians is greater than that from hunter gatherers. This suggests that extensive migration of post-Neolithic agropastoralists with larger population sizes is likely to have occurred in other regions of the Old World.

Finally, this hypothesis should cause us to expect maximal bottleneck intensity just before the founding population of Neolithic males began to expand—in other words, just before the initial Neolithic. However, the bottleneck inferred from the data peaks 1 to 2 millennia after the initial Neolithic in every region of the Old World. The bottleneck should also have ended as Neolithic colonization of each continental region was completed, unlike what is reflected in the temporal trajectory of the inferred bottleneck.

The third hypothesis is that increase in intragroup social and material inequalities associated with the Neolithic Transition increases male reproductive variance, as well as the transmission of such variance across generations. Karmin et al.[4] proposed that changed social structures of post-Neolithic populations, contributing to increased reproductive variance, could account for the bottleneck. Indeed, agropastoral cultures display more heritable wealth and social hierarchy, which may increase male reproductive variance[28–30], and polygyny is more prevalent than among hunter gatherers[31]. At the same time, patrilineality could interact with social and material inequality to increase heritability of male reproductive success along father–son genealogies[32]. Increased male reproductive variance and the transmission of such variance could reduce Y-chromosomal diversity in post-

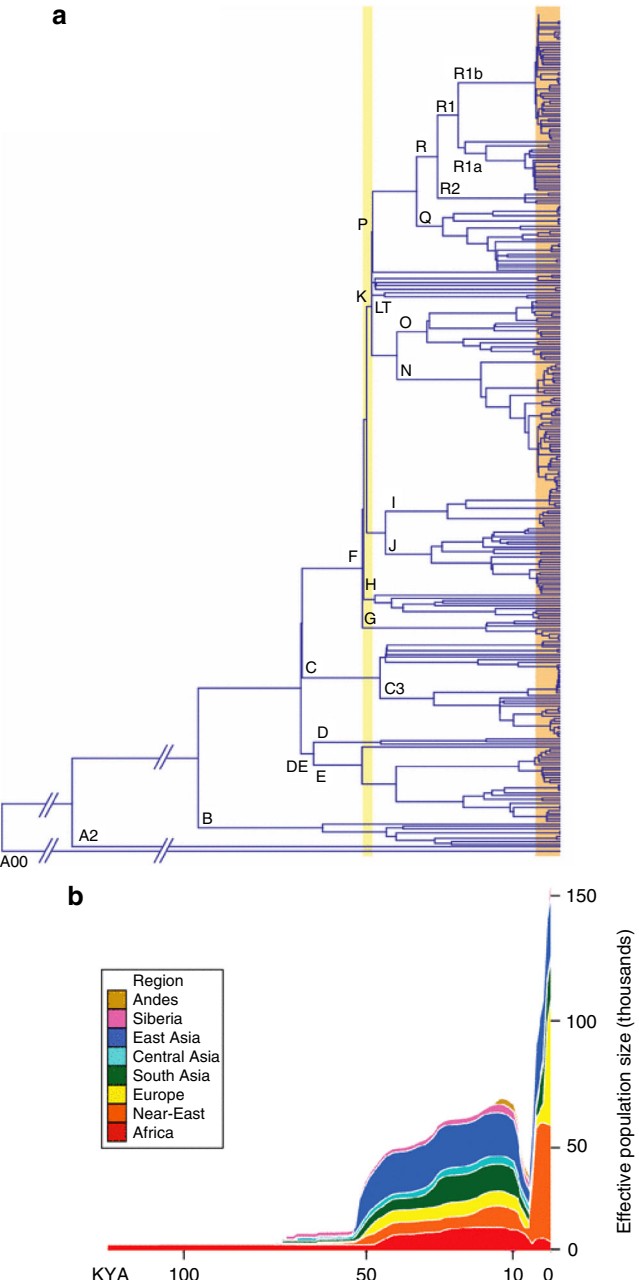

**Fig. 3** Male-specific Y (MSY) chromosome phylogeny from next-generation sequencing data, and associated demographic reconstruction. **a** MSY phylogeny based on 456 samples and 35,700 SNPs. Major haplogroups are labelled. The orange box highlights recent expansions identified in several haplogroups, and the yellow box highlights more ancient expansion of deep-rooting lineages. **b** MSY Bayesian Skyline Plots (of effective population size against time), with different world regions indicated by colours as shown in the key. Reprinted from Batini and Jobling[78] with permission from Mark A. Jobling and Springer Science +Business Media

Neolithic cultures[33,34] and could account to a certain degree for the bottleneck.

However, several pieces of evidence suggest that the roles that intragroup reproductive inequality, and transmission of such inequality play in the bottleneck are relatively minor. First, the effects of social status on male reproductive success in hunter gatherers, agriculturalists and pastoralists are of similar magnitude[35]. Second, a reproductive ratio of 1:17 falls far outside ratios

typical in ethnographic data[36]. Such extreme skews are inconsistent with the social dynamics of anti-dominance coalitions in small-scale human groups[37]. Third, long-term transmission of reproductive success, causing the steady accretion of reproductive advantage to certain patrilineages, may circumvent these issues. However, as social inequality contributes to the strength of hereditary transmission of reproductive success, and social inequality increases and does not decline with the rise of political complexity and social stratification that coincides with the end of the bottleneck period, changes in the transmissibility of reproductive success could not be the major factor causing the bottleneck.

In other words, social hierarchy and differences in material wealth show an increasing trend as one enters the Late Neolithic, up to the emergence of political complexity in the Old World[38], and we should therefore expect the bottleneck to increase in intensity, with the rising impact of wealth and social status on reproductive success in large-scale, stratified societies[39]. However, the bottleneck lifts in each part of the Old World during precisely the period that coincides with the rise of regional polities, chiefdoms and states. Therefore, trends in material and social inequality, which may increase male reproductive variance, do not appear to track changes in bottleneck intensity, and some other factor must have been responsible for the depression of Y-chromosomal diversity during the bottleneck period.

Bringing together anthropological theory, mathematical models and archaeological and archaeogenetic findings, we show how the formation of patrilineal kin groups and intergroup competition among these groups, which disproportionately affected males, could have led to a reduction in Y-chromosomal diversity that was much greater than the reduction in male population size, while keeping the female population size relatively stable. We suggest that unlike the mitochondrial DNA phylogeny, the Y-chromosomal phylogeny contained large numbers of clades that went extinct within the bottleneck period due to sociocultural processes. These clades, if they had been maintained within the population, would have caused coalescent-based algorithms, such as BEAST, to show parallel and congruent population dynamics for Y-chromosomes and mtDNA.

## Results

**Patrilineal kin group competition explains bottleneck**. Our hypothesis explains the bottleneck as a consequence of intergroup competition between patrilineal kin groups, which caused cultural hitchiking between Y-chromosomes and cultural groups and reduction in Y-chromosomal diversity. Competition between demes can dramatically reduce genetic diversity within a population[1], especially if the population is structured such that variation is greater between demes than within demes. Culturally transmitted kinship ideals and norms can cause homophilous sorting and limit interdemic gene flow, creating homogeneous demes that differ strongly from one another. Patrilineal corporate kin groups, with coresiding male group members descending from a common male ancestor, would produce such an effect on Y-chromosomes only, as patrilineal corporate kin groups generally coexist with female exogamy[40], which would homogenize the mitochondrial gene pools of different groups[41,42].

With intergroup competition between patrilineal corporate kin groups, two mechanisms would operate to reduce Y-chromosomal diversity. First, patrilineal corporate kin groups produce high levels of Y-chromosomal homogeneity within each social group due to common descent, as well as high levels of between-group variation. Second, the presence of such groups results in violent intergroup competition preferentially taking place between members of male descent groups, instead of between unrelated individuals. Casualties from intergroup

competition then tend to cluster among related males, and group extinction is effectively the extinction of lineages.

The first mechanism, leading to strong correlations between clades of Y-chromosomes and sociocultural groups, produces population structure through a culturally transmitted ideal. The second mechanism describes how such structure interacts with competitive social dynamics to affect the genetics of populations.

If the primary unit of sociopolitical competition is the patrilineal corporate kin group, deaths from intergroup competition, whether in feuds or open warfare, are not randomly distributed, but tend to cluster on the genealogical tree of males. In other words, cultural factors cause biases in the usually random process of transmission of Y-chromosomes, increasing the rate of loss of Y-chromosomal lineages and accelerating genetic drift. Extinction of whole patrilineal groups with common descent would translate to the loss of clades of Y-chromosomes. Furthermore, as success in intergroup competition is associated with group size, borne out empirically in wars[43] as 'increasing returns at all scales'[44], and as larger group size may even be associated with increased conflict initiation, borne out in data on feuds[45], there may have been positive returns to lineage size. This would accelerate the loss of minor lineages and promote the spread of major ones, further increasing the speed of genetic drift.

In addition, the assimilation of women from groups that are disrupted or extirpated through intergroup competition into remaining groups is a common result of warfare in small-scale societies[46]. This, together with female exogamy, would tend to limit the impact of intergroup competition to Y-chromosomes.

In social terms, as patrilineages are repeatedly lost over time and extant sociocultural groups trace their cultural descent to fewer and fewer progenitors, Y-chromosomal variation would be reduced. The Y-chromosomal clades that survived into the post-bottleneck period would tend to demonstrate sparser branching during the bottleneck period itself, in contrast to the branching rates of the mtDNA phylogeny, as large numbers of clades that existed or were created during the bottleneck period would have become extinct during the bottleneck. This would bias methods used to estimate effective population size (i.e., the number of individuals in a population who contribute offspring to the next generation), because an episode of strongly reduced diversity would give the appearance of a population bottleneck, even in the absence of fluctuations in demography.

This hypothesis has an added benefit in that it could explain the temporal placement of the bottleneck if competition between patrilineal kin groups was the main form of intergroup competition for a limited episode of time after the Neolithic transition. Anthropologists have repeatedly noted that the political salience of unilineal descent groups is greatest in societies of 'intermediate social scale' (Korotayev[47] and its citations on p. 2), which tend to be post-Neolithic small-scale societies that are acephalous, i.e. without hierarchical institutions[48]. Corporate kin groups tend to be absent altogether among mobile hunter gatherers with few defensible resource sites or little property (Kelly[49] pp. 64–73), or in societies utilizing relatively unoccupied and under-exploited resource landscapes (Earle and Johnson[50] pp. 157–171). Once they emerge, complex societies, such as chiefdoms and states, tend to supervene the patrilineal kin group as the unit of intergroup competition, and while they may not eradicate them altogether as sub-polity-level social identities, warfare between such kin groups is suppressed very effectively[51,52]. These factors restrict the social phenomena responsible for the bottleneck to the period after the initial Neolithic but before the emergence of complex societies, which would place the bottleneck-generating mechanisms in the right period of time for each region of the Old World.

To explore our proposal quantitatively, we formulate and analyse two mathematical models, adopting both analytical and computational approaches. Our analytical approach is based on the Lotka–Volterra model, a nonlinear dynamical process describing the dynamics of populations constrained by cooperative or competitive relations. For our computational approach, we propose a computational grid model, which allows simultaneous tracking of the dynamics of cultural groups and haplogroups. We defer detailed description of the mathematical models to Methods, and mathematical proofs to Supplementary Note 1. Supplementary Notes 5 and 6 discuss various other possible models, including their limitations and potential applicability.

We prove that, in a Lotka–Volterra model with two male subpopulations and a subpopulation of females shared between both male subpopulations, competition between the male subpopulations drives one of the male subpopulations to extinction. If each male subpopulation represents a genetically homogeneous patrilineal descent group, then the model corresponds to the second mechanism in which extinction of a group due to competition leads to eradication of an entire male descent group. Detailed information on our analytical model can be found in Methods, and our proof in Supplementary Note 1.

To model the impact of our proposed mechanisms more realistically with diverse populations and varying levels of patrilineality, we construct a discrete-time computational grid model. This model includes intergroup competition, group extinction and group fission, and models their effects on haplogroup diversity in males given different levels of patrilineality, in a population divided into sociocultural groups that are equivalent to 'tribes'. The model is comprised of a male population subdivided into cells in a grid of cultural groups, or 'tribes' (rows), and haplogroups (columns). With each 'generation', or step in the simulation, deaths from intergroup competition between cultural groups, mutation, reproduction and sometimes group fission occurs (Fig. 4). Initial conditions can either be patrilineal (PT), in which case the Y-chromosome carried by any given individual is perfectly correlated with his cultural group identity, or non-patrilineal (NPT), in which case they are perfectly uncorrelated, as seen in Fig. 4a. Additionally in the reproduction step, we ensure that population growth in each generation after deaths from intergroup competition restores population size to its initial magnitude. This serves as a constraint on our model's performance in generating bottleneck-like phenomena, showing that it may be able to generate changes in diversity (if it actually does so) in the complete absence of fluctuations in male population size, and, by implication, independently of extreme sex ratios and polygamy. More detailed information on our computational grid model may be found in Methods.

Starting with equal frequencies of all haplogroups, 18 sets of parameters based on different combinations of values of the three parameters listed in Table 1 were used for our simulations (c.f., Supplementary Table 1). We ran our simulations for 60 generations, or 1.5 millennia, approximating the length of the bottleneck. Detailed results for all 18 simulations can be found in Supplementary Figs. 2–19 under Supplementary Note 3. We observe a consistent pattern across all parametrizations. Simulations of societies structured by patrilineal kin groups display increasing divergence in haplogroup frequencies over time, and a large number of haplogroups go extinct fairly early in the simulation, while one or several haplogroups rapidly become overwhelmingly dominant in frequency. Simulations without patrilineality display different frequency dynamics, with few or no haplogroups going extinct, and all haplogroups approximately equally represented even at the end of the simulation.

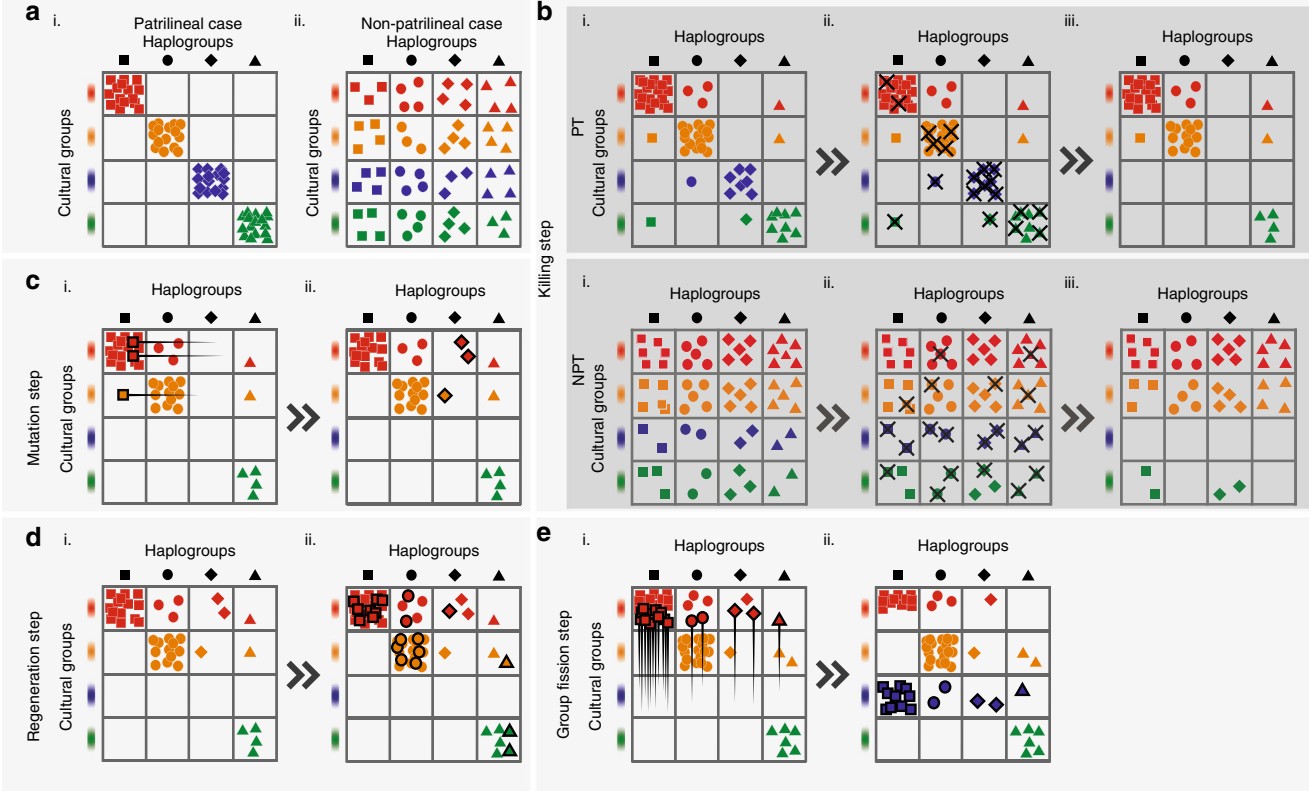

**Fig. 4** Schematic of the steps in the simulation, according to the order described in the algorithm. **a** (i) Patrilineal (PT) starting conditions, where cultural groups strictly determine haplogroup type. **a** (ii) The non-patrilineal (NPT) condition where they are perfectly uncorrelated. **b** The killing step, with a more (PT) and less (NPT) patrilineal starting condition. The number of deaths in each group is inversely related to group size. The blue cultural group goes extinct in both cases. This causes the haplogroup represented by the diamonds to go extinct in PT, but no haplogroup extinction occurs in NPT. **c** The mutation step, where a small number of individuals in the largest haplogroup change their haplogroup. **d** The regeneration step, where (i) is a replica of (**b**) PT (iii), and (**d**) (ii) shows how the original number of individuals before the killing step is restored by proportionally increasing the number of individuals in all cells. **e** Group fission step. Where an empty row occurs, the largest cultural group splits, and half the individuals form a new cultural group in the empty row. The step in which we remove cultural groups that are too small—between (**c**, **d**) (see Methods)—is not shown

**Table 1 Parameters and model choices used in the simulations[a]**

|  | Description | Setting choices |
| --- | --- | --- |
| *Parameter* |  |  |
| Total population size, or $N^{total}$ | Total number of individuals in the simulation | 10,000, 20,000, 30,000 |
| Death rate | Intensity of intergroup competition, proxied by the percentage population decline in the first cultural group at the beginning of the simulation | 15%, 25%, 50% |
| Patrilineality | Correlation between cultural group identity and haplogroup type among individuals in the simulation | Patrilineal, non-patrilineal |
| *Model modifications* |  |  |
| Group extinction rule | The treatment of cultural groups which fall below 20 individuals in the simulation | Extirpation (all individuals in group killed), fusion (individuals in the group join another cultural group in row above or below in the grid) |
| Presence of Cultural Selection | Whether or not different cultural groups have different reproductive fitness due to differences in culture | Fitness differential present, no fitness differential present |
| \|C\|:\|H\| ratio | Number of different cultural groups and haplogroups possible in the simulations, i.e. number of rows vs number of columns in the simulation grid | 1:5, 1:25 |

[a]Simulation results are presented in Supplementary Note 3 for all setting choices for the first three parameters, which characterize the 18 sets of simulations run on our original, unmodified model. The last three parameters—headed by 'model modifications' to distinguish them from the first three—are possible modifications to our main model. The details and results of simulations run using these modified models can be found in Supplementary Note 4

Our model offers two striking parallels to the Y-chromosome dynamics that were empirically observed by Karmin et al.[4] and Poznik et al.[6]. First, simulations with patrilineality are characterized by large reductions in diversity, reproducing a

bottleneck-like phenomenon in the Y-chromosome even in the absence of fluctuations in male population size across generations. In other words, our model shows that intergroup competition between patrilineal kin groups can lead to large

losses of diversity in Y-chromosomal haplogroups despite near-constant male demographic size over 1.5 millennia. Thus, the simulations show that our proposed sociocultural mechanism can reproduce conditions under which a population bottleneck may be inferred over the timescale observed in Karmin et al.[4]

Second, while all haplogroups initially have equal frequencies, in all cases with patrilineality, one or several haplogroups eventually dominate. These dominant haplogroups rapidly increase in frequency in the first half of the simulation. A rapid, sustained increase in the frequency of a particular Y-chromosomal haplogroup over a short timespan may cause the rapid, proportional accumulation of mutations among carriers of that clade over that timespan, due simply to numerical and proportional advantage. This may match the conditions under which star-shaped 'bursts' appeared in the Y-chromosome genealogy of dominant clades among modern populations[6].

These two characteristics distinguish simulations with (PT) and without (NPT) patrilineality, and are robust to a wide range of sociocultural parameters, as seen in the preservation of this distinction in all 18 of our simulations. A representative example of the differences in dynamics between patrilineal and non-patrilineal conditions is shown in Fig. 5. These distinctions persist even after modifications to our model, such as the incorporation of cultural selection and fusion of cultural groups (see Supplementary Note 4 for details). Therefore, our model reliably produces dynamics that strongly and plausibly replicate two features of the population genetics of the Y-chromosome in that period: the reduction in diversity, interpreted as a population bottleneck[4], and the occurrence of star-shaped phylogenies[6]. This supports our hypothesis that patrilineal kinship may cause cultural hitchhiking between Y-chromosomes and sociocultural groups, leading to altered population dynamics that ultimately leave their mark on genetic diversity.

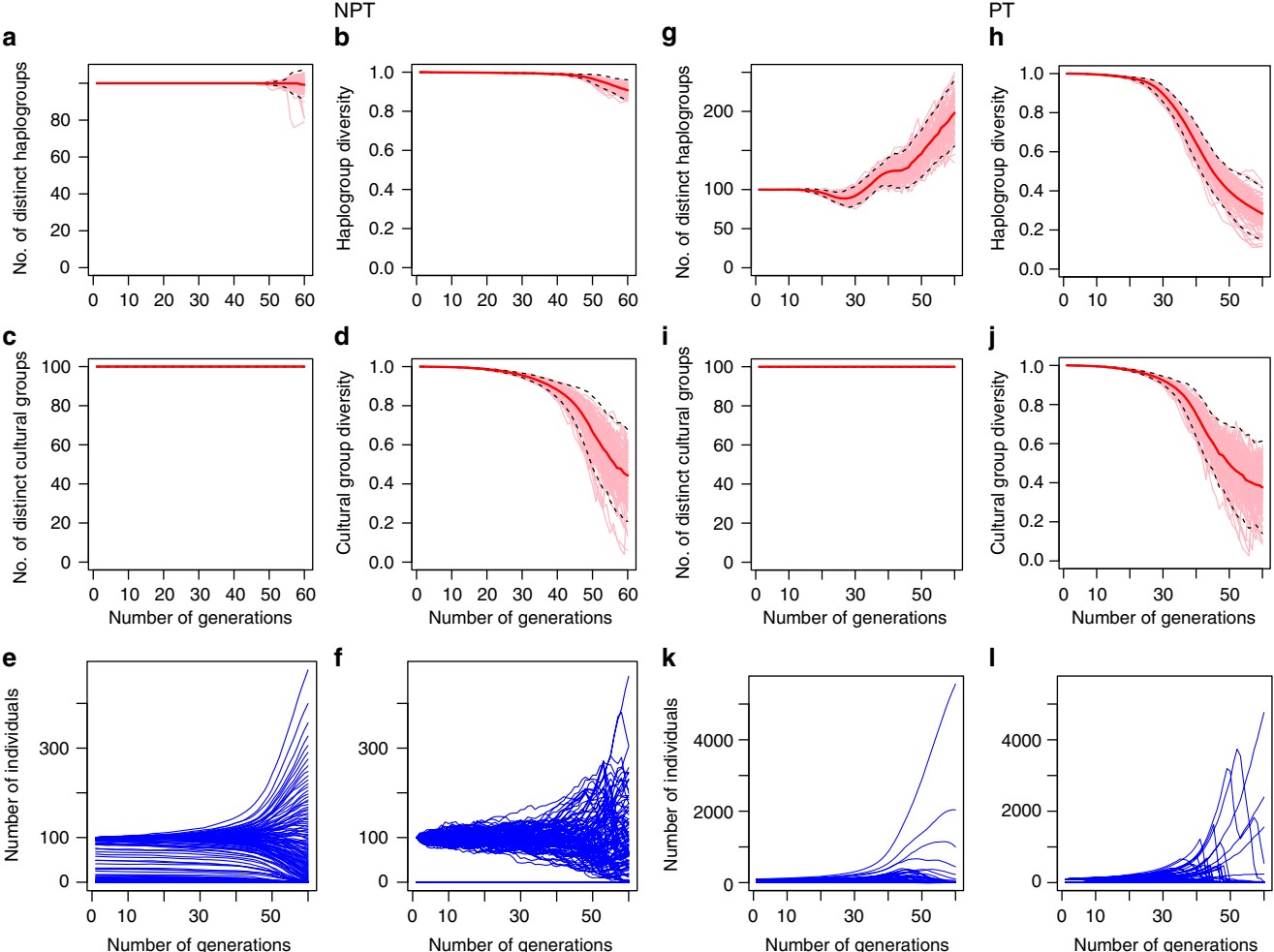

**Fig. 5** Cultural group and haplogroup dynamics in competing completely non-patrilineal groups (NPT) and in competing patrilineal kin groups (PT). These correspond to Figs. 2 and 3 of Supplementary Note 3. There are 100 cultural groups, or 'tribes,' each with size 100 at the beginning, and 500 possible haplogroups. Dynamics of the number of distinct cultural groups, cultural group diversity, the number of distinct haplogroups and haplogroup diversity are obtained by averaging over 100 simulation runs (details in Supplementary Note 3). Left panels: **a** Number of distinct haplogroups. **b** Haplogroup diversity. **c** Number of distinct cultural groups. **d** Cultural group diversity. Each graph consists of 100 pink trajectories representing 100 distinct simulation runs, and one thick solid red trajectory representing the average of the 100 pink trajectories. Black dotted lines representing confidence intervals covering two standard deviations from this mean trajectory. Note that the total number of haplogroups does not increase in this model as each haplogroup-cultural group pair has too few representatives for the mutation step of our algorithm to operate on; this is a methodological artifact and not a significant result. **e** The mean dynamics of the size of each of the 500 haplogroups over $T = 60$ generations. **f** The dynamics of each of the 500 haplogroups for one single run. Right panel: **g–j** Dynamics for the patrilineal configuration, correspond to **a–d** of the non-patrilineal configuration. Similarly, **k**, **l** correspond to **e**, **f**

## Discussion

We point to three caveats with respect to our computational model. In our model, mutation changes the haplogroup identity of individuals. This is not a good representation of real Y-chromosome mutations, which only change the haplotype of Y-chromosomes. Haplogroups, on the other hand, are clades of related Y-chromosome haplotypes defined post-hoc by researchers through the naming of nodes on a tree; the concept of a 'haplogroup mutation' has no counterpart in real life. However, we face computational limitations when attempting to model the dynamics of haplotypes, which are extremely diverse and are shared by at most several individuals out of a population of millions. We utilize the intuition that only a subset of the mutations occurring within the ancient population we are simulating lead to clades of Y-chromosomes that become named as haplogroups by researchers who observe the phylogenetic tree at a later date. To implement this, we use a 'haplogroup mutation rate', or the rate at which haplogroup-defining mutations occur, which is much lower in our model than the Y-haplotype mutation rates found in the literature. We discuss approaches that may resolve this discrepancy in Supplementary Note 6.

A second caveat for our model is that patrilineality is treated exogenously. It is treated as a preexisting distinction between two sets of social ecologies whose origins are left unexplained. While the origin of unilineal descent groups is difficult to model, anthropologists have suggested various hypotheses, ranging from social adaptation to the presence of heritable wealth and accumulations of property[49], or cultural evolution under conditions of intergroup competition[48]. Modelling the evolution of the causative social structures could form the basis for an extension of the current work.

The third caveat concerns the possibility that the cultural groups exhibit spatial structure, which our model does not address. Stepping stone dynamics for both competition and group fusion, which is a more realistic representation of population ecology, should result in haplogroup similarity being correlated with geographical distance between cultural groups. Supplementary Note 4 includes a 'stepping stone'-type fusion of cultural groups as a modification of our model.

Our proposal is supported by findings in archaeogenetics and anthropological theory. First, our proposal involves an episode in human prehistory when patrilineal descent groups were the socially salient and major unit of intergroup competition, bracketed on either side by periods when this was not the case. That such a sequence of sociocultural scenarios is plausible is supported by the extensive literature on the unilineal kin group in anthropology, which emphasizes its prevalence in societies with 'mid-range complexity,' and its absence or weakness in both the simplest and most complex societies ('curvilinear trend'[53], also see citations on page 2 of Korotayev[47]). Cross-cultural data suggests that unilineal kin groups became widespread with the increase in population density and ecological intensification associated with increasing reliance on agriculture, animal husbandry, or stationary resource concentrations (see Johnson and Earle[50] pp. 45–51, 157–171), only to decline in importance or even disappear entirely with the emergence of complex regional polities, which create 'conditions unfavourable for large unilineal kinship groups'[53].

Indeed, Kelly[49] finds that segmentary societies, or societies where 'social substitutability' of members of descent groups was apparent during conflicts, were generally small-scale societies with frequent warfare, and that such societies were either agropastoralists or sedentary hunter gatherers with food storage or exploiting stationary resource concentrations. Nomadic hunter gatherers with low population densities were not among this group. Ember et al.[48], using a data from the Ethnographic Atlas,

show that unilineal descent groups were always present in societies with endemic warfare but without centralized authority —i.e. in any non-state non-chiefdom society where war was present. Johnson and Earle[50] and Kelly[49] suggest that corporate kin groups, as means of social organization and as agents in intergroup conflict, may arise with increasing density, intensity of ecological exploitation, importance of capital investment and social circumscription, all factors which tend to increase in an agropastoralist landscape, and which also increase the importance of enforceable and excludable property rights. Intensive agriculture and pastoralism, by increasing the need for enforcement of social rights to resources, may have led individuals to agglomerate and incorporate along the most constant and least overlapping set of interpersonal ties, those of unilineal kinship. Indeed, unilineal descent groups are highly efficient at mobilizing for collective action, including intergroup competition[54,55]. Thus, the patrilineal corporate kin group may have become more prevalent due to its utility in competition and in guaranteeing access to vital resources after the intensification of environmental exploitation during the Neolithic transition.

On the other hand, the development of political complexity in chiefdoms and states tends to reduce the prominence of corporate kin groups. They may not entirely disappear, but their relevance as units of mobilization in intergroup competition must be reduced if sustained increases in social scale were to be achieved[44,56]. Increased market involvement under state peace also tends to catalyse the eventual breakup of descent groups in cultures with long histories of state rule[47]. Trajectories of political development in Madagascar[57] and Polynesia[58] provide some insight into the process by which kinship structures were steadily superseded as major units of violent competition following the emergence of the state or regional polity.

There is evidence that other analogous situations involving gene-culture hitchhiking in culturally-defined social groups may have affected genetic diversity. Central Asian pastoralists, who are organized into patriclans, have high levels of intergroup competition and demonstrate ethnolinguistic and population-genetic turnover down into the historical period[59]. They also have a markedly lower diversity in Y-chromosomal lineages than nearby agriculturalists[42,60]. In fact, Central Asians are the only population whose male effective population size has not recovered from the post-Neolithic bottleneck; it remains disproportionately reduced, compared to female estimates using mtDNA[4]. Central Asians are also the only population to have star-shaped expansions of Y-chromosomes within the historical period, which may be due to competitive processes that led to the disproportionate political success of certain patrilineal clans[60].

Analogous phenomena are also found among nonhuman species. In whales, species with matrilineal organization such as Orcas, Pilot and Sperm whales have reduced mitochondrial diversity. The reduction is such that cultural selection operating at the group level[61,62] has been invoked to explain the phenomenon.

Another line of evidence supporting our proposal comes from the shallow coalescence observed in phylogenies constructed from ancient DNA samples found in Europe. Archaeogenetic sequencing of samples from diverse cultures in Europe has enabled investigation of inter-population relationships[11,63]. Less investigated is the pattern of internal genetic relations between members of the same cultural group.

We use shallowness of coalescence, a qualitative feature of phylogenies, to describe how often coalescences occur in the tree of Y-chromosome sequences from archaeogenetic samples from single cultures. In cultures with deep coalescence, Y-chromosomes coalesce at very long timescales; in cultures with shallow coalescences, many men carry Y-chromosomes that

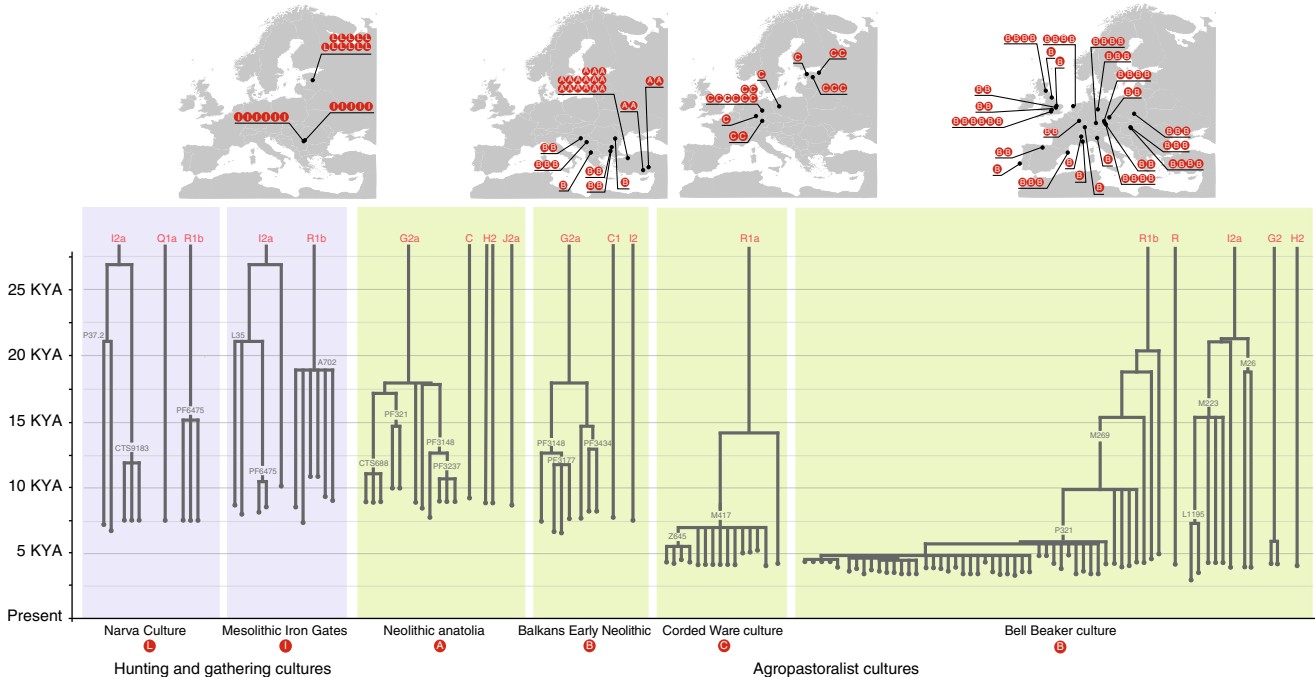

**Fig. 6** Tree of Y-chromosome genotypes from samples found among cultures with hunter-gatherer subsistence, and agropastoralist subsistence. The blue background represents hunter-gatherer subsistence while the green background represents agropastoralist subsistence. Letters in red circles match individuals from sites with their archaeological context. Note that R1b-P321 is synonymous with R1b-S116. Adapted from Figs. 3, 4, 5 and 6 of Kivisild[67], with addition of information from Olalde et al.[64]. The vertical axis represents time; the position of branch points represent the ages of branch-defining mutations, with nomenclature and age from yfull (https://www.yfull.com/tree/)

coalesce a short time prior to their deaths. Cultures with deeper coalescences contain males with low levels of paternal relatedness; the converse applies for cultures with shallow coalescences. Figure 6 displays Y-chromosomal phylogenies of multiple samples from across Europe from cultures with both hunter-gatherer and agropastoral modes of subsistence.

Figure 6 shows a striking pattern of differences in shallowness of coalescence in samples from hunter-gatherer, farmer and pastoralist cultures. While hunter-gatherer Y-chromosomes from the same culture, and often the same sites, commonly divide into haplotypes that coalesce in multiple millennia, Y-chromosomes of samples from farmer and pastoralist cultures are more homogeneous and have more recent coalescences. The Bell Beaker culture has a high proportion of sampled males (81%) from a large geographical area (Iberia to Hungary) who belong to an identical Y-chromosomal haplogroup (R1b-S116), implying common descent from a kin group that existed quite recently. Some groups of males share even more recent descent, on the order of ten generations or fewer[64]. Such recent common descent may even be retained in cultural memory via oral genealogies, such as among descent groups in Northern and Western Africa, whose members can trace descent relationships up to three to four centuries before the generation currently living[40]. Likewise, from Germany to Estonia, the Y-chromosomes of all Corded Ware individuals sampled, except one, belong to a single clade within haplogroup R1a (R1a-M417) and appear to coalesce shortly before sample deposition.

Thus, groups of males in European post-Neolithic agropastoralist cultures appear to descend patrilineally from a comparatively smaller number of progenitors when compared to hunter gatherers, and this pattern is especially pronounced among pastoralists. Our hypothesis would predict that post-Neolithic societies, despite their larger population size, have difficulty retaining ancestral diversity of Y-chromosomes due to

mechanisms that accelerate their genetic drift, which is certainly in accord with the data. The tendency of pastoralist cultures to show the lowest Y-chromosomal diversity and the shallowest coalescence would also be explained, as they may have experienced the social conditions that characterized cultures of the Central Asian steppes[42]. Indeed, the Corded Ware pastoralists may have been organized into segmentary lineages[65], an extremely common tribal system among pastoralist cultures, including those of historical Central Asia[66].

Additionally, the homogeneity in Y-chromosomes across the geographic spread of the extensive Bell Beaker and Corded Ware cultures may testify to continuity in patrilineal kinship even with group fission and long-distance migration, which may be consistent with a phenomenon observed in modern ethnographies of patrilineal segmentary cultures, that of 'predatory expansions'[54].

The rate of coalescence of Y-chromosomal haplogroups may not be fully captured by currently available Y-chromosomal genotyping, as most archaeogenetic samples are sequenced through hybridization capture methods, not whole-genome sequencing[67]. Such methods enrich for human-specific DNA in SNP (single-nucleotide polymorphism) arrays targeted towards phylogenetically relevant SNP sites in modern populations, which allows for the capture of phylogenetic information from samples with low endogenous DNA content. These methods tend to lose information from chromosomal regions far from SNP sites in the array, which may cause downstream mutations private to the ancient samples that unite them at phylogenetically more recent nodes to be missed. Therefore, Fig. 6 is only a preliminary demonstration of patterns in genetic data, and may change with data from ancient Y-chromosomes sequenced at higher resolutions.

Our hypothesis represents an attempt to synthesize genetic, anthropological and archaeogenetic data to create a synoptic view of social dynamics, or the social process, as a braided stream in

time and space. The social mechanisms we describe in our hypothesis may have caused co-transmission of Y-chromosomes in social groups with developments in social structure after the Neolithic, causing an episode of massively increased gene-culture hitchhiking[61] that affected only Y-chromosomes, which then ended with the emergence of complex societies. Our mathematical models support our hypothesis and illustrate the utility of combining data from genetics and archaeogenetics with theoretical models that approximate realistic social and anthropological scenarios. This represents a step forward in synthesizing the growing body of information emerging from ancient DNA studies. Indeed, future modelling approaches (see Supplementary Notes 5 and 6), together with the excavation and sequencing of greater numbers of archaeogenetic samples, will continue to put our hypothesis to the test.

Exploration of uniparental variation in modern human populations from cultures at different social scales or with different political histories, or from some geographic areas that have societies that never left the 'bottleneck period' (such as the Central Asian pastoralists from Karmin et al.), or from societies that never entered it, may represent further tests of our hypothesis. Some findings give us preliminary indications about the impact of historical social structure on genetic variation. For example, Irish men sharing the same surname, even very common ones, are approximately 30 times more likely to share a Y-chromosomal haplotype than a random pair of Irish men[68,69]. This is not the case for the English, who demonstrate much weaker signals of coancestry between Y-chromosomes and surnames, especially when the surname is very common[70,71]. This difference parallels contrasts in the relative importance of patrilineal clans in the social origins of surnames in the history of the two populations.

There are also significant variations in bottleneck intensity, with the bottleneck being less extreme in East Asian and Southeast Asian populations than in West Asian, European, or South Asian populations (see Fig. S4a, b of Karmin et al.[4]). The wider distribution and greater importance of pastoral cultures in the ancient Middle East, Europe and India may have played a role in creating this difference. This may imply that prehistoric interactions between farmers and herders in the Middle East and South Asia involved interesting social patterns, as in Europe. Farmer-herder interactions have been recognized as playing a key role in cultural change, such as in the creation of ideals of personal, mobile and alienable property[72], or the rapid development of political complexity[38].

At the same time, shallowness of coalescence may be useful in anthropology and archaeology, as it may allow for inference about social and political realities from archaeogenetic sequences. It may be used as a measure of intensity of intergroup competition and rates of social group extinction, if we have sufficient evidence that the cultures under investigation were patrilineal and patrilocal. Conversely, if the cultures were matrilineal and matrilocal, intense warfare may have produced negligible effects on Y-chromosomal diversity. The impact of warfare on mitochondrial diversity in matrilineal societies represents an interesting research question. Archaeogenetic sequences from around the world, including from societies that were inferred to be matrilineal, could be profitably investigated for insights into past social conditions.

Lastly, multiple characterizations of the 'chiefdom'—conceived as a grade of political and social development—have emerged in the literature[73], and our hypothesis has interesting implications for the interaction between kin groups and emerging social complexity. Many theorists assert that the transition from kinship-based to non-kinship-based mechanisms of sociopolitical organization was a critical episode in human social evolution[44,56,74]. The emergence of ideologies that help to legitimise and propagate

a social class which cuts across kin groups would, in these theories, constitute a crucial juncture in the emergence of 'ultra-societies' in which kinship is no longer the most important element in sustaining sociopolitical relations and coalition formation. Ideologies that support legitimate power, and the institutional positions or offices that possess such power, may now exist as a parallel set of cultural elements that, alongside kinship, also enable the creation of sociopolitical structures. As 'conquest war' replaces 'competition war'[50]—a transition in accord with our hypothesis as well—the institutions and social formations created by such ideologies, as well as the ideologies themselves, may serve as the new target of cultural selection, instead of the populations involved in the competitive process.

If all ultrasocieties are the products of historical junctures where kinship organizations were supervened through cultural and ideological change, this should be supported by differences in patterns of uniparental genetic variation between cultural groups with different political histories. An investigation into the patterns of uniparental variation among, for example, the Betsileo highlanders of Madagascar, who may have undergone an entry and an exit from the 'bottleneck period' very recently[57], could reveal phenomena relevant to such history. Cultural changes in political and social organization—phenomena that are unique to human beings—may extend their reach into patterns of genetic variation in ways yet to be discovered.

## Methods

**Modified Lotka–Volterra model.** The Lotka–Volterra equations are widely used in mathematical modelling of biochemical, ecological and economic systems. The flexibility of this system comes from the ease of modelling either competitive or cooperative interactions between distinct species or populations. In our case, we focus on the competition model[75], in which we assume that populations of males are competing with one another. However, we modify the standard model by including an exogenous population of females whose reproductive capacity affects the growth rate of each male population. This model attempts to simulate a social scenario with male populations competing against each other, where the total male population size is constrained by the size of the female population, and the relative population replenishment of each male group after intergroup competition in each generation is proportional to their relative sizes. In social terms, this translates to the simplifying assumption that migration, assimilation or capture of females between groups produces equal female availability in each group, and therefore the number of male births in the next generation is proportional to the current number of males.

A special case of our model with just two male populations competing against one another has the following form.

$$dY_1(t) = Y_1(t)(r_1 X(t) - c_{11} Y_1(t) - c_{12} Y_2(t))dt$$
$$dY_2(t) = Y_2(t)(r_2 X(t) - c_{21} Y_1(t) - c_{22} Y_2(t))dt \qquad (1)$$
$$dX(t) = aX(t)\left(1 - \frac{X(t)}{K}\right)dt,$$

where the two male populations, with sizes $Y_1(t)$ and $Y_2(t)$, compete against one another, alongside an exogenous female population with size $X(t)$. Note that $a$, $K$, $c_{ij}$, and $r_i$ are positive constants. It turns out that under Eq. (1) with $\min(c_{12}, c_{21}) > \max(c_{11}, c_{22})$, almost always one population of males takes over the other. See Supplementary Note 1 for a proof, and Supplementary Note 5 for more on the Lotka–Volterra approach (notably, stochastic modifications).

Thus in a large population consisting of two male subpopulations and a subpopulation of females shared between both male subpopulations, competition between the male subpopulations drives one of the male subpopulations to extinction. If each male subpopulation represents a genetically homogeneous patrilineal descent group, then the model corresponds to our second mechanism in which extinction of a group due to competition leads to eradication of an entire male descent group.

Under a standard Lotka–Volterra system consisting of two competing populations, if interpopulational competition dominates intrapopulational competition, eventually one of the two competing populations dies out[75]. This observation is generalizable to higher dimensional Lotka–Volterra competition models, where the competitive exclusion principle states that if $n \geqslant 2$ male populations depend on $m < n$ resources (following Hofbauer and Sigmund[75], pp. 47–48, in our case, $m = 1$ and the resource is the female population), then at least one of the $n$ populations will go extinct. Like the standard Lotka–Volterra system, our modified Lotka–Volterra model reflects male group extinction under intergroup competition, which may be expected to reduce Y-chromosomal diversity over time.

**Computational grid model**. To capture the impact of our proposed mechanisms in more realistic environments comprising diverse populations and varying levels of patrilineality, we construct a discrete-time computational grid model. At one extreme, we define a society with no patrilineal or patrilocal tendencies to consist of sociocultural groups or 'tribes' with average intragroup male relatedness no different from that of males from different groups, i.e. completely non-patrilineal (NPT) groups. This would be reflected in our model through Y-chromosomal haplogroup distributions in each group identical to that of the population as a whole. At the other extreme, societies structured completely by patrilineal (PT) kin groups, would comprise sociocultural groups or 'tribes' within which Y-chromosomal haplogroups are identified by virtue of strict and complete common descent. Our model also includes mutation, which increases diversity over time, and group splitting, which keeps the size of each 'tribe' below a reasonable threshold, and maintains the total number of sociocultural groups. We further specify that when the population of a 'tribe' falls below a certain fixed threshold, it goes extinct, which corresponds to the social extinction of a long tail of groups with unsustainably small numbers of members.

We investigate the effect of intergroup competition over 60 generations, on the order of 1.5 millenia, given estimates of human generation time[6], on Y-chromosomal diversity at different levels of patrilineality, and under a range of realistic anthropological and archaeological parameters, such as total population size and group size. The total population size is again kept constant over time, in order to investigate the magnitude of any 'bottleneck' effects that our mechanisms can produce in the absence of real demographic fluctuations.

In this model, a large population $N^{total}$ of males is partitioned into a number, $|C|$, of cultural groups, and a number, $|H|$, of distinct haplogroups. Competition between cultural groups and mutation of haplogroups occur at each generation, but the total population size is kept constant. The processes that constitute one step of our simulation, i.e. one generation, are represented pictorially in Fig. 4. (The simulation details can be found in Supplementary Note 3.) During the competition step (Fig. 4b), individuals of haplogroup $h$ from cultural group $c$, i.e. from the cultural group-haplogroup pair $(c, h)$ (where $c \in C$, $h \in H$), are killed. The number of individuals killed follows a Poisson probability distribution, where the proportion of each group killed is greater for smaller groups. This reflects the strong effect of group size on relative success in human intergroup competition[76]. During the mutation phase (Fig. 4c), within cultural group $c$ a proportion of individuals from the major haplogroup—that is, the haplogroup $h_c$ for which the number of individuals belonging to the pair $(c, h_c)$ is maximum amongst all $(c, h)$—mutate into different haplogroups (they still remain within the same cultural group). We further assume that whenever a cultural group size drops below 20, that group is no longer viable and loses all its members; this prevents the persistence of cultural groups with a single or very few members, which does not conform to anthropological reality. To ensure the total population size is constant we scale the $|C| \times |H|$ array of cultural group-haplogroup pairs $(c, h)$ upwards such that the new generation of males has population size of $N^{total}$. In other words, the number of individuals in each haplogroup-cultural group increases proportionally to the number already present, such that the starting population size is recovered (Fig. 4d). Finally, to control the growth of cultural group sizes and to maintain the overall number of cultural groups at a constant level, whenever a cultural group goes extinct, the cultural group with the largest size splits in half with one half remaining in the original cultural group and the other half forming a new cultural group (Fig. 4e). This parallels the fusion-fission dynamics of sociocultural groups in small-scale human societies. It resembles but differs from the Maruyama–Kimura model[77], where extinction of subpopulations or demes is determined by a static parameter $\lambda$ instead of by dynamic intergroup competition, and each mutation gives rise to a new allele (analogous to a haplogroup in our treatment) under an infinite alleles framework.

Our model includes multiple tuning parameters and variables, and the steps described above follow the order described. We provide a complete algorithmic specification in Supplementary Note 2. The components of the model and choices of parametrisation are described in Table 1. Details of the parameters and simulations results are summarized in Supplementary Table 1. We specify the number of generations $T$, to run the steps described above, for the computational grid model of the competition and mutation dynamics.

We note the model itself can also be modified, in ways listed in Table 1, to incorporate cultural selection and group fusions. We discuss our findings with these modifications in Supplementary Note 4.

**Code availability**. The simulation code for our computational grid model is available online at https://github.com/alanaw1/CulturalHitchhiking.

**Data availability**. All data supporting the findings are found in the paper and in its Supplementary Notes. Unprocessed simulation outputs are available from the authors upon request.

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

## Acknowledgements

We thank Ken Aoki for reading the manuscript; Carlos Bustamante, Mark Jobling, Monika Karmin, and David Poznik for permission to reuse figures; and Hilla Behar, William Gilpin, Volker Heyd, Oren Kolodny, Kristian Kristiansen, Philip Labo, Amaury Lambert, Julia Palacios, Noah Rosenberg, Peter Turchin and Peter Underhill for helpful comments, discussions and suggestions. This research was supported in part by the Center for Computational, Evolutionary and Human Genomics (CEHG), the Morrison Institute for Population and Resource Studies at Stanford and NSF Grant BCS-1515127 to Noah Rosenberg.

## Author contributions

T.C.Z., A.A and M.W.F. designed the project. T.C.Z. and A.A. did the analysis, and T.C.Z., A.A. and M.W.F. wrote the paper.

## Additional information

**Competing interests:** The authors declare no competing interests.

