## [Peer Review File · Nature Communications]

Reviewers' comments:

Reviewer #1 (Remarks to the Author):

Overall, I find the study to be an interesting exploration and application of novel and potentially game-changing concepts previously introduced by gene-culture coevolutionary researchers such as Boyd, Richerson, Whitehead, and Premo. I think the study is a worthwhile contribution—the conclusions follow logically from the results, and the authors introduce an interesting and unique potential explanation for an intriguing empirical observation. I provide my line-by-line comments and suggestions below in the hope that they might help the authors strengthen what is already an interesting and solid paper.

Line-by-line comments/questions/suggestions

Page 1: "In human populations, changes in genetic variation are driven not only by genetic processes, but can also arise from cultural or social changes (Premo and Hublin, 2009)."

Yes, indeed, this is important. A more recent paper by Premo provides an accessible review of this issue and probably should be cited here along with Premo and Hublin.

Premo L. S. 2012. Hitchhiker's guide to genetic diversity in socially structured populations. *Current Zoology* 58:287-297.

Page 5: "If the primary unit of socio-political competition is the patrilineal corporate kin group, deaths from intergroup competition, whether in feuds, raids or open warfare, are not randomly distributed, but tend to cluster on the genealogical tree of males."

Yes, well stated. The authors make it clear that the important factor here is not necessarily the number of male deaths but rather how the deaths are distributed among different male lineages within a socially structured population. This is really the lynchpin of the paper, and I think the authors do a good job of getting this point across and supporting it.

Page 5: "Genetic drift is no longer random, increasing the rate of loss of Y-chromosomal lineages."

The authors could benefit by providing another sentence or two here to explain the concept of "non-random genetic drift" for non-specialists who may not be familiar with that concept.

Page 6: "This would bias methods used to estimate effective population size because an episode of reduced diversity would give the appearance of a population bottleneck, even in the absence of fluctuations in demography."

Yes, indeed! This is a very good point that is largely underappreciated by geneticists and anthropologists who work with human data. I wonder if a very brief but clear definition of "effective population size" is in order here? I think it could be helpful, as that is a topic of confusion among the archaeological and anthropological crowd.

Page 8: "We further specify that when the population of a "tribe" falls below a certain fixed threshold, it goes extinct, which corresponds to the social extinction of a long tail of groups with unsustainably small numbers of members."

Well, that is one way to do it. But another way would be to let these stragglers join/fuse with an adjacent (or randomly chosen) group. My intuition tells me this is probably closer to how most anthropologists would model the scenario. Regardless of that, I wonder how these two different assumptions about what happens with very small groups would affect the results of the simulation. It might be worthwhile to rerun the simulations under the assumption described above and see if the same amount of reduction is achieved by inter-lineage competition. If not, then the results are particularly sensitive to this seemingly benign assumption.

Page 9.

So, a group fission occurs only after a group extinction event makes a grid cell available? I think this is a fine way to model the effects of local group extinction, but it is not clear to me that this is the best representation of "the fusion-fission dynamics of sociocultural groups in small-scale human societies." Perhaps there should be a brief discussion as to how the effect of patrilinearity might differ under a different assumption of group replacement.

And, more generally, when discussing the effects of local group extinction and recolonization on genetic diversity, it is worthwhile to cite this study:

Maruyama, T. and Kimura, M. 1980. Genetic variability and effective population size when local extinction and recolonization of subpopulations are frequent. *Proc. Natl. Acad. Sci. USA* 77: 6710–6714.

"Our model includes multiple tuning parameters and variables"

I admit that this raises a bit of a red flag. Why are these "multiple parameters" not discussed at greater length in the main text? Were sensitivity analyses run on all of these "tuning" parameters? I had a quick look at the supplementary material and this information did not pop out at me there either, which is a bit concerning. Perhaps I just missed it.

I agree that the following 2 major claims are supported by their simulation results:
"Thus, the simulations show that our proposed sociocultural mechanism can produce reduced diversity over the timescale observed in Karmin et al. (2015), reproducing conditions under which a population bottleneck may be inferred."

"A rapid, sustained increase in frequency of a particular Y-chromosomal haplogroup over a short timespan may cause the rapid accumulation of mutations among carriers of that particularly frequent clade within that timespan, due simply to numerical advantage. This may match the conditions under which star-shaped 'bursts' appeared in the Y chromosome genealogy of dominant clades among modern populations."

Page 10.

I understand that the grid based model is not spatially explicit. How would the dynamics be affected if empty cells were populated by the largest *adjacent* group rather than the largest group in the metapopulation? I would predict that patrilinearity would have less of an effect on diversity, but perhaps the effect is not attenuated as much as one might suspect at first blush. In any event, given that these populations were situated in space, this is a caveat worthy of some discussion in this paper.

"There are two caveats we can point to with respect to our models."

Just two? Surely there are more caveats to discuss in this case. The effect of space is just one more that would be of interest to the readers. I'm sure there are 4-5 others that could be mentioned, if not discussed in detail.

Page 12: "The reduction is such that cultural selection operating at the group level (Whitehead, 1998) may need to be invoked to explain the phenomenon."

This is a good statement. I would recommend citing a very recent paper by Whitehead and colleagues here in addition to Whitehead's 1998 paper:

Whitehead, H., F. Vachon, and T. R. Frasier. 2017. Cultural Hitchhiking in Matrilineal Whales. *Behav Genet* 47:324-334.

Section 5.3 is unnecessary to the author's argument, and I would recommend that it is removed from the paper. At the very least, section 5.3 should be moved to the supplementary material. Removing this section opens up more space for addressing some of the more pressing issues raised above.

Page 14: "flatness of descent"

I get this concept and I think it is worth discussing in the case of the ancient genetic record of Europe, but I'm not sure it is necessary to introduce the new term "flatness of descent" in

doing so. Does this term really provide more traction or additional information than the notion of a shallow coalescent? I'm not sure that it does, but I could be wrong. If the authors think they can make the same point using more familiar terms, then I would recommend using those terms without introducing this one.

Page 16: "Exploration of uniparental variation in modern human populations from cultures at different social scales or with different political histories with very dense sampling may serve as a "synchronic test" of our hypothesis. In other words, that some geographic areas may have societies which never left the "bottleneck period" (such as Central Asian pastoralists from Karmin et al.), while other geographic areas may have societies that never entered it, may further confirm or disconfirm the applicability of our hypothesis."

Nice. This is a particularly cool idea for an empirical test of the processes investigated here.

Page 17: "Flatness of descent may be used as a measure of intensity of intergroup competition and rates of social group extinction, if we have sufficient evidence that the cultures in question were patrilineally organised and practiced patrilocal post-marital residence."

Agreed, BUT what would be "sufficient evidence" be in this case, especially for societies for which we only have archaeological evidence? Perhaps a few more sentences here would be helpful for those who would like to develop such tests.

Page 27: Figure 4.

The two larger, blue panels on the right of Fig. 4 depict lines that remain at 0 from generation 0 to 60. Do these lines represent haplotypes that were not displayed in the population at the start of the simulation? If so, is there a need to display these haplotypes on these panels at all? It seems like these lines that remain at 0 might cause more confusion than they alleviate. Related question: why doesn't mutation introduce new haplotypes through time? Should some of the haplotypes not present at generation 1 appear (though at low frequencies) throughout the simulation? Does figure 4 not include mutation? If the sims did include mutation, can mutation introduce a haplotype that was not displayed by at least one individual at the start? Perhaps I missed this detail when reading through it. But if not, then this could stand to be clarified.

Reviewer #2 (Remarks to the Author):

This manuscript examines the effect of kinship and social structures on human genetic diversity through mathematical models. Specifically, it attempts to find an explanation to

the enigmatic male specific bottleneck dating to the last 5-7KYA that had been revealed by recent studies comparing Y chromosome and mtDNA diversity of human populations from across different continents. The key novelty of the work lies in the proposal of the model of competition between patrilineal corporate kinship groups to explain the sex-specific patterns in empirical data. This proposal relies on the synthesis of wide range of anthropological, archaeological and genetic evidence in light of the results of computer simulation results generated by the authors. The manuscript is well and clearly written, the topic should be of general interest to the wide readership of the journal. I cannot identify any major flaws in the arguments while I think that the balance of the contextual and novel material presented in the main text should be shifted towards the latter. At the moment only one of the main figures represents novel findings supporting the proposed model. While the contextual evidence is important I think it can be presented in much more condensed form. As for the competing models that could explain the bottleneck I also found that more clarity should be given to the distinction of the proposed competition between patrilineal kinship groups and the patrilocality case, is the latter part of the proposed model or can the authors effectively rule out that patrilocality with patrilineality alone without the competition between the corporate patrilineal groups could not explain the findings.

Reviewer #3 (Remarks to the Author):

see attached file

In their paper CULTURAL HITCHHIKING: COMPETITION BETWEEN PATRILINEAL KIN GROUPS AND THE POST-NEOLITHIC Y-CHROMOSOME BOTTLENECK (TIAN CHEN ZENG; ALAN J. AW2;, AND MARCUS W. FELDMAN), authors present an interesting work with a mathematic development regarding patrilineal kin that could explain the discrepancy between Y chromosome and mt DNA diversity in Eurasia.

I have some major comments:

Zeng et al propose a model based on competition between patrilineal kin groups to explain Y chromosome genetic patterns in human populations. While such proposition is interesting, other socio-cultural processes may reduce the Y chromosome diversity and the authors should refer more extensively to previous literature on this subject. For example, Smouse (genetics 1981) showed that a dynamics of lineal fissions may reduce greatly the effective size of the population, and Heyer and colleagues studied patrilineal populations in Central Asia and showed that the Y chromosome genetic diversity in these patrilineal (and patrilocal) populations is greatly reduced in comparison to cognatic (and patrilocal) populations, suggesting a strong impact of the patrilineal dynamics on these sex-specific genetic patterns. In addition, Heyer et al showed that the transmission of reproductive success is greater in patrilineal than in cognatic populations which may contribute to the reduction of Y-chromosome diversity in these populations. They developed several population genetics statistics (see Chaix et al, 2004; Chaix et al 2007(current Biol) Marchi et al 2017; Heyer et al 2016 et al, (and ref in the papers) in relation with patrilinearity with a special focus on imbalance of coalescence tree.

In particular, transmission of reproductive success may explain the particular shape of the coalescent tree described in Zeng et al paper (see figure 5 “tree of Y chromosomes” and Heyer et al, TIG).

In other words, a dynamics of lineal fissions with extinctions of kin groups (linked to demographic stochasticity) and/or the transmission of reproductive success may be sufficient to lead to an important Y chromosome diversity reduction.

Of course alternative models can explain this shape of coalescent tree and are worth testing which is the aim of this paper. But in the introduction these alternative processes that may shape Y chromosome diversity should be clearly stated.

Further, authors should also explain how to test between alternative models.

On the general theory behind the model developed by Zeng et al:

- One of the hypotheses behind the model is that there is competition/war between lineages. There is, to my knowledge (except for old school social anthropologists) no evidence that patrilineal populations are more prone to war than non patrilineal populations. Indeed base on anthropological theory the contrary is expected: patrilineality increases exogamous behaviors; exogamy is mainly used to build up alliances and therefore to reduce war/competition (Levi-Strauss' theory). Further, do we have any archeological evidence of such an increase of war/death when this social system is thought to have started in Europe?
- In their model, women are chosen at random (they constitute a “common resource”). Although I understand that a model is always a simplification of reality, random mating is clearly not the case in patrilineal populations. Conversely, there is a strong tendency to have non-random mating with matrilineal kin and 41% of unilineal populations prefer to marry with a first degree cousin.

- *“On the other hand, the development of political complexity in chiefdoms and states tends to reduce the prominence of corporate kin groups. They may not entirely have disappeared, but their salience as social identities and relevance as units of mobilization in political competition must have been reduced if sustained increases in social scale were to be achieved (Richerson and Christiansen (2013), also see chapter 2 of Yooe (2005)). Increased market integration under state peace also tended to catalyze the eventual breakup of descent groups in cultures with long histories of state rule (Korotayev, 2003).”*
This is clearly an “old fashion” vision of social systems and too much in an “evolution” perspective with a direction in the evolution of social systems. Indeed, society can be unilineal with complex political systems.

More technical problem in relation with the model:

- o Women (as a common resource) are incorporated in the “simple” Lotka-Volterra approach. However the population dynamics of women being solely considered as dependent on a growth rate and a carrying capacity that are independent of the dynamics of the other (male) populations, it does not seem to have any impact on the total dynamics. It seems that replacing the female dynamics by constant K would yield the same results, make the same point, and render the understanding easier to fathom.
- o For non-specialists of the L-V competition equation, a phase portrait for the model with two or three male populations, would make it clearer to readers where the equilibria are located, whether they are stable or unstable and that some male populations will go extinct for any possible initial population.
- o Given that in the more “complex” computational grid model, the number of females is considered constant, why not use the same parameter for the simpler LV model ?
- It seems that having the parameter of the Poisson law of killed individuals indexed on $\sqrt{N_{c,h}(t)}$ has a strong effect on the dynamics. . Why using root square? (Is this an algorithmically-driven or anthropologically-driven relationship). To which extend do the result depend on this parameter? Given this choice for the more “complex” computational grid model, why not use the same parameter for “simple” LV model (in which competition is related to $N_c(t)$ and not $\sqrt{N_c(t)}$)
 - o I may be mistaken, but it seems that another issue with indexing on the square root, is that the number of individuals killed in clan c, is $K_c(t) = \sum_h K_{c,h}(t) = Poi[\alpha_c, \rho, [\sum_{c' \neq c} N_{c',h}(t)] [\sum_h \sqrt{N_{c,h}(t)}]]$ which contrary to the case where the indexation is on $N_{c,h}(t)$, is NOT the same r.v. as $Poi[\alpha_c, \rho, [\sum_{c' \neq c} N_{c',h}(t)] [\sqrt{\sum_h N_{c,h}(t)}]]$.
 - o And so, to my mind, contrary to what is stated on page 5 of the SI, I am under the impression that “The mean proportion killed per generation of a group c is NOT proportional to the product of the total number of individuals not belonging to c and the square root of the number of individuals belonging to c ». Could you clarify this point ?
- Mutation: the process call “mutation” does not create here new mutation. This is a simple move of individuals from one haplogroup to the other. This is very intriguing.
- Mutation does not create new haplogroups but consists in the translation of individuals from one haplogroup to all others. If we understand correctly, whilst the set of all possible

haplogroups/haplotypes is fixed, most of these sets are empty at first (300/400). What is the effect of the ratio of $\frac{H}{C}$ (which is 4-fold in your model) on the resulting dynamics?

- Mutation only affect the “major haplogroup”. Again, is this an algorithmically driven choice ? What would be the effect of allowing all mutations for each haplogroups ?
- Why do you assume that the fitness of a cultural group is linearly related to its group number? There is already advantage based on the size of the group, since death rate is proportionate to the square of the size of the group. (and again, why the square of the group size?). What are the dynamics in a model where all cultural groups have the same fitness, but with the square root effect favoring larger groups ?
- Weeding: what is the logic to kill all groups that have less than 20 individuals? What is the importance of that threshold ?
- Replenishment: why half of the group is moved to a new grid cell ? The anthropological/ecological motivations for this portion of the algorithms are not clear

Other major comments:

The paper is too long, the part on Madagascar (although interesting) is out of scope of the paper. Idem for the part on Polynesia. It would need to be the object of another paper with genetic data.

Flatness of descent: other population genetics statistics have already been developed in regards with Y-chromosome genetic diversity in patrilineal population (Chaix et al Curr Biol 2007, Marchi et al, AJPA 2017). More work is needed to propose and validate a new population genetic statistics. I although guess that what you call flatness of descend is simply a ratio of coalescent branch length. How it compares to other statistics that have already been developed to infer patrilineality?

Minor comments on the models:

In the presentation of the LV model (in the main text, p.7), the condition on the competition parameters for extinction of one population is expressed as “ $c_{12}, c_{21} > c_{11}, c_{22}$ ”. Is this equivalent to $\min(c_{12}, c_{21}) > \max(c_{11}, c_{22})$. If it is the case, the latter expression is easier to understand. However then, how does this mathematical condition differ from the litteral condition “if interpopulation competition dominates intrapopulation competition” expressed a few lines later. It seems to be equivalent, and this should (if it is the case) be made clearer.

The reference to the “competitive exclusion principle” of Hofbauer and Sigmund would benefit from further explanation on the role of women as “resources” and why the other populations (males popualtions) do not constitute “ressources”. I suppose it is related to the domain of definition of parameters c_{ij} , but this would make it easier to understand were that point to be clarified.

In the algorithm description of the “computational grid model”:

- Is it correct that in step 3.ii it is N_{c,h_c} that is reduced by the mutation ? shouldn't it be $N_{c,h_c}^{unscaled}$?
- In step 6 (replenishment), it is not clear what happens in the case of several groups c having $\sum_h N_{c,h}(t+1) = 0$.

Figure 2,3,4,5 supp material.

One point is not clear: figure A show that some haplogroups disappear (for example in fig2, in some case the value is lower than 100) therefore I would expect in figure on the right to see some haplogroupe reaching a value of zero.

Correct haplotypes in figure/haplogroups in legend (figure 3)

Author's Replies and Revisions

I. Reviewer #1: Comments to the Authors

Overall, I find the study to be an interesting exploration and application of novel and potentially game-changing concepts previously introduced by gene-culture coevolutionary researchers such as Boyd, Richerson, Whitehead, and Premo. I think the study is a worthwhile contribution—the conclusions follow logically from the results, and the authors introduce an interesting and unique potential explanation for an intriguing empirical observation. I provide my line-by-line comments and suggestions below in the hope that they might help the authors strengthen what is already an interesting and solid paper.

Authors' reply: Thank you for your detailed comments, which have been very helpful indeed, and your positive appraisal of our paper.

Line-by-line comments/questions/suggestions

- Page 1: "In human populations, changes in genetic variation are driven not only by genetic processes, but can also arise from cultural or social changes (Premo and Hublin, 2009)."

Yes, indeed, this is important. A more recent paper by Premo provides an accessible review of this issue and probably should be cited here along with Premo and Hublin. Premo L. S. 2012. Hitchhiker's guide to genetic diversity in socially structured populations. *Current Zoology* 58:287-297.

Authors' reply: We agree, and have cited this paper in line 8.

"In human populations, changes in genetic variation are driven not only by genetic processes, but can also arise from cultural or social changes (Premo and Hublin, 2009; Premo, 2012)." In our quotes from the paper, we have italicised any changes we have made.

- Page 5: "If the primary unit of socio-political competition is the patrilineal corporate kin group, deaths from intergroup competition, whether in feuds, raids or open warfare, are not randomly distributed, but tend to cluster on the genealogical tree of males."

Yes, well stated. The authors make it clear that the important factor here is not necessarily the number of male deaths but rather how the deaths are distributed among different male lineages within a socially structured population. This is really the lynchpin of the paper, and I think the authors do a good job of getting this point across and supporting it.

- Page 5: "Genetic drift is no longer random, increasing the rate of loss of Y-chromosomal lineages."

The authors could benefit by providing another sentence or two here to explain the concept of "non-random genetic drift" for non-specialists who may not be familiar with that concept.

Authors' reply: We agree, and have added a sentence for clarification (line 197):

"If the primary unit of socio-political competition is the patrilineal corporate kin group, deaths from intergroup competition, whether in feuds, raids or open warfare, are not randomly distributed, but tend to cluster on the genealogical tree of males. *Cultural factors bias the randomness intrinsic to the transmission of Y chromosomes*, increasing the rate of loss of Y-chromosomal lineages."

- Page 6: “This would bias methods used to estimate effective population size because an episode of reduced diversity would give the appearance of a population bottleneck, even in the absence of fluctuations in demography.”

Yes, indeed! This is a very good point that is largely underappreciated by geneticists and anthropologists who work with human data. I wonder if a very brief but clear definition of “effective population size” is in order here? I think it could be helpful, as that is a topic of confusion among the archaeological and anthropological crowd.

Authors' reply: We agree, and have added a definition:

“This would bias methods used to estimate effective population size, namely the number of individuals in a population who contribute offspring to the next generation, because an episode of reduced diversity would give the appearance of a population bottleneck, even in the absence of fluctuations in demography.”

- Page 8: “We further specify that when the population of a “tribe” falls below a certain fixed threshold, it goes extinct, which corresponds to the social extinction of a long tail of groups with unsustainably small numbers of members.”

Well, that is one way to do it. But another way would be to let these stragglers join/fuse with an adjacent (or randomly chosen) group. My intuition tells me this is probably closer to how most anthropologists would model the scenario. Regardless of that, I wonder how these two different assumptions about what happens with very small groups would affect the results of the simulation. It might be worthwhile to rerun the simulations under the assumption described above and see if the same amount of reduction is achieved by inter-lineage competition. If not, then the results are particularly sensitive to this seemingly benign assumption.

We agree that group extinction via fusion is a possibility we should consider. To explore the effect of such fusion suggested by the reviewer, we have performed a series of simulations where group fusion occurs when group sizes fall below the 20-person threshold, and this co-occurs with group fission to maintain a steady state in group number and prevent runaway effects on group size. We observe no changes in the qualitative outcome of the simulations. Patrilineal configurations lead to the death of most lineages and the dominance of few (< 5) major lineages, much as before. The relevant figures in the Supplementary Materials, C21, C23 and C25 show that including fusion in the model has no substantive effect.

- Page 9.
So, a group fission occurs only after a group extinction event makes a grid cell available? I think this is a fine way to model the effects of local group extinction, but it is not clear to me that this is the best representation of “the fusion-fission dynamics of sociocultural groups in small-scale human societies.” Perhaps there should be a brief discussion as to how the effect of patrilinearity might differ under a different assumption of group replacement.

Authors' reply: We agree that this is a weakness of our proposal.

At the same time, part of the reason why we specified the model in this way is because we do not want to introduce directional dynamics in group size and group number into the simulation, which would occur if fusion took place between groups when some of them become too small. We wish to assume a steady-state for group size and cultural group number (cultural diversity), which appears to us to be a good baseline assumption. Allowing steady directional change in group sizes and group number may result in a better fit to real-world phenomena (as complex societies with millions of individuals do in fact arise in the historical record) but would add many complications to our model. For example, social structure also changes along with social scale, as diverse kinship groups are incorporated into one socio-political unit. Instead, we treat

separately the situation of large-scale societies with diverse kinship groups using a separate category of simulations, the non-patrilineal simulations. We do this because the relevant unit of intergroup competition between large-scale societies is not the kinship group, but the socio-political unit with members of many kinship groups, and this is the aspect of theoretical relevance here.

And, more generally, when discussing the effects of local group extinction and recolonization on genetic diversity, it is worthwhile to cite this study:

Maruyama, T. and Kimura, M. 1980. Genetic variability and effective population size when local extinction and recolonization of subpopulations are frequent. Proc. Natl. Acad. Sci. USA 77: 6710–6714.

Authors' reply: We agree, and have cited this paper in line 345.

- “Our model includes multiple tuning parameters and variables”

I admit that this raises a bit of a red flag. Why are these “multiple parameters” not discussed at greater length in the main text? Were sensitivity analyses run on all of these “tuning” parameters? I had a quick look at the supplementary material and this information did not pop out at me there either, which is a bit concerning. Perhaps I just missed it.

We agree with your comments. In addition to changing our language on line 404 (“*We point to three caveats*”), we have, after taking into consideration both your and other reviewer’s comments, divided the parameters that could be expected to have an effect on our model into three classes, and accounted for their effects. We refer to them as “**hyperparameters**” to distinguish them from the 18 sets of parameters validating model robustness across varying population sizes and competition rates among patrilineal and nonpatrilineal configurations. These hyperparameters are:

1. The cultural group-haplogroup (C:H) ratios --- currently set at 1:5
2. The presence of a fitness differential, in the form of a linear law --- amounting to the presence of selection amongst cultural groups
3. The “killing off” process: whether fusion or fission

To keep our model assumptions in line with the sociocultural process we describe, we have removed the fitness differential linear law, thereby eliminating effects of selection. We call this model the “**null model**” to avoid confusion in the language. We find no changes in the qualitative behaviour of cultural group and haplogroup dynamics. We report simulations for this process across the same 18 different parametrizations listed on Table 1 in Section 3 of the supplement.

Moreover, we consider the sensitivity of the dynamics of our process to the three hyperparameters, by tweaking each of them and performing simulations. The results of this sensitivity analysis are summarised in the following five points:

1. By changing the cultural group-haplogroup (C:H) ratios, we observe no changes in the qualitative outcome of the simulations: patrilineal configurations lead to the death of most lineages and the dominance of few (< 5) major lineages.
2. By adding culture-related fitness differentials (i.e., reverting to the previous model), we find that there is quicker spreading out of various haplogroup lineages, and the lineages eventually vanish or persist in the same qualitative manner as they do in our null model simulations.
3. As mentioned above in our reply to your comment about fusion, we observe no changes in the qualitative outcome of the simulations when we change the outcome of cultural

group extinction from complete removal of individuals to fusion with another cultural group.

4. By both adding culture-related fitness differentials and changing the group extinction rule to fusion, we find that, compared with applying point 3 alone, the haplogroup lineages spread out more quickly, but eventually die or persist in the same qualitative manner. Moreover, the variance in the stochastic dynamics for haplogroup diversity is reduced.
5. From points 2 and 4, we deduce that selection (1) speeds up the spreading out of the winning and losing haplogroup lineages, and (2) reduces the variation in the diversity dynamics.

The manipulation of the three hyperparameters of our model could potentially form the basis for a test to measure whether and how selection or fusion rules affect specific aspects of the haplogroup dynamics of competition among cultural groups, for instance the speed of divergence of dominant lineages. We point this out in section C.1 of the supplementary materials.

- I agree that the following 2 major claims are supported by their simulation results:
“Thus, the simulations show that our proposed sociocultural mechanism can produce reduced diversity over the timescale observed in Karmin et al. (2015), reproducing conditions under which a population bottleneck may be inferred.”
“A rapid, sustained increase in frequency of a particular Y-chromosomal haplogroup over a short timespan may cause the rapid accumulation of mutations among carriers of that particularly frequent clade within that timespan, due simply to numerical advantage. This may match the conditions under which star-shaped ‘bursts’ appeared in the Y chromosome genealogy of dominant clades among modern populations.”

Authors' reply: We appreciate your positive appraisal.

- Page 10.
I understand that the grid based model is not spatially explicit. How would the dynamics be affected if empty cells were populated by the largest *adjacent* group rather than the largest group in the metapopulation? I would predict that patrilinearity would have less of an effect on diversity, but perhaps the effect is not attenuated as much as one might suspect at first blush. In any event, given that these populations were situated in space, this is a caveat worthy of some discussion in this paper.

Authors' reply: We agree with your comments. We did not include spatial distribution in our model as it is already somewhat complex and including spatial separation of cultural groups may make the model rather intractable. We may deal with this issue in future papers.

- “There are two caveats we can point to with respect to our models.”

Just two? Surely there are more caveats to discuss in this case. The effect of space is just one more that would be of interest to the readers. I'm sure there are 4-5 others that could be mentioned, if not discussed in detail.

Authors' reply: We agree with your comment, and have changed our language on line 404 (“We point to three caveats”). We have also discussed the impact of modelling important parameters in different ways, which we indicate above in our reply on your comment about tuning parameters.

- Page 12: “The reduction is such that cultural selection operating at the group level (Whitehead, 1998) may need to be invoked to explain the phenomenon.”

This is a good statement. I would recommend citing a very recent paper by Whitehead and colleagues here in addition to Whitehead's 1998 paper:

Whitehead, H., F. Vachon, and T. R. Frasier. 2017. Cultural Hitchhiking in Matrilineal Whales. *Behav Genet* 47:324–334.

Authors' reply: We agree, and have cited this paper in line 513.

- Section 5.3 is unnecessary to the author's argument, and I would recommend that it is removed from the paper. At the very least, section 5.3 should be moved to the supplementary material. Removing this section opens up more space for addressing some of the more pressing issues raised above.

Authors' reply: We agree. We have reduced this section to a single sentence (line 489-492) that just makes the point that evidence does exist from Madagascar and Polynesia that kinship groups are superseded as the relevant unit of intergroup competition after the transition to large-scale societies.

- Page 14: "flatness of descent"
I get this concept and I think it is worth discussing in the case of the ancient genetic record of Europe, but I'm not sure it is necessary to introduce the new term "flatness of descent" in doing so. Does this term really provide more traction or additional information than the notion of a shallow coalescent? I'm not sure that it does, but I could be wrong. If the authors think they can make the same point using more familiar terms, then I would recommend using those terms without introducing this one.

Authors' reply: We agree with your comment, and have changed the language in section 5.3 (references to "flatness of descent" becomes "*shallow coalescence*") to reflect this. We are also doing ongoing work to quantify "shallow coalescence," as well as what it would mean comparatively in human datasets (relative to what?), as well as the implications we can draw from it about social structure.

- Page 16: "Exploration of uniparental variation in modern human populations from cultures at different social scales or with different political histories with very dense sampling may serve as a "synchronic test" of our hypothesis. In other words, that some geographic areas may have societies which never left the "bottleneck period" (such as Central Asian pastoralists from Karmin et al.), while other geographic areas may have societies that never entered it, may further confirm or disconfirm the applicability of our hypothesis."

Nice. This is a particularly cool idea for an empirical test of the processes investigated here.

Authors' reply: We appreciate your positive comments.

- Page 17: "Flatness of descent may be used as a measure of intensity of intergroup competition and rates of social group extinction, if we have sufficient evidence that the cultures in question were patrilineally organised and practiced patrilocal post-marital residence."

Agreed, BUT what would be "sufficient evidence" be in this case, especially for societies for which we only have archaeological evidence? Perhaps a few more sentences here would be helpful for those who would like to develop such tests.

Authors' reply: There is skeletal isotope data sufficient to demonstrate strong patrilocality, for example, in the LinearBandKeramik early Neolithic culture in Europe:

- Bentley, R. Alexander, et al. "Community differentiation and kinship among Europe's first farmers." *Proceedings of the National Academy of Sciences* 109.24 (2012): 9326-9330.

- Bentley, A. L. E. X. "Mobility, specialisation and community diversity in the Linearbandkeramik: isotopic evidence from the skeletons." *Proceedings-British Academy*. Vol. 144. Oxford University Press Inc.
- Bentley, R. Alexander. "Mobility and the diversity of Early Neolithic lives: isotopic evidence from skeletons." *Journal of Anthropological Archaeology* 32.3 (2013): 303-312.nc., 2007.

If such strong patrilocality in settlements is also correlated with strong continuity and homogeneity in Y-chromosomal lineages of burials within settlements, a hypothesis of patrilineally organised sociopolitical units may be supported. Such a hypothesis may, for example, be proved for the earliest Anatolian farmers by the BEAN (Bridging the European and Anatolian Neolithic) project, which aims to sequence large numbers of samples from many sites, as well as obtain isotope data.

We did not include this information in the paper as we felt it was of insufficient relevance.

- Page 27: Figure 4.
The two larger, blue panels on the right of Fig. 4 depict lines that remain at 0 from generation 0 to 60. Do these lines represent haplotypes that were not displayed in the population at the start of the simulation? If so, is there a need to display these haplotypes on these panels at all? It seems like these lines that remain at 0 might cause more confusion than they alleviate. Related question: why doesn't mutation introduce new haplotypes through time? Should some of the haplotypes not present at generation 1 appear (though at low frequencies) throughout the simulation? Does figure 4 not include mutation? If the sims did include mutation, can mutation introduce a haplotype that was not displayed by at least one individual at the start? Perhaps I missed this detail when reading through it. But if not, then this could stand to be clarified.

Thank you for pointing out the potential confusion arising from printing size 0 haplogroup lineages. As a matter of fact these lineages start at 0, but may increase due to chance mutation events in our model. It appears that some lineages do not increase significantly from an initial number of 0. However, we think that displaying these lineages captures the haplogroup dynamics in its entirety, and is necessary for the sake of accurate representation of our model. (That is, there is a non-zero chance that a haplogroup that starts out at 0 becomes non-zero by the end of the run due to mutation, even though for some runs they stay at 0 throughout. We want this feature to be captured in the graph.)

Mutation does introduce new haplogroups through time. However, there is only a finite number of possible haplogroups to mutate to, a consequence of computational constraint. We do consider modifications to our model in which the number of possible haplogroups is increased from 500 to 2500. Because we start out with 100 haplogroups in the beginning, the remaining 2400 haplogroups are new haplogroups that individuals can mutate to.

Figure 4 (now Figure 5) presents simulations of our algorithm, which has a mutation step. In our algorithm, a mutation can, and in fact typically does, introduce haplogroups that were not displayed by a single individual at the start.

A potential source of confusion is the apparent absence of an increase in the number of distinct haplogroups over time for simulations under a non-patrilineal configuration over 100 runs. This is a result of having too few individuals in each haplogroup-cultural group cell of our grid under the non-patrilineal condition, in conjunction with the low mutation rate. Note that there are some runs for which the number of distinct haplogroups increases by the end of 60 generations; this is captured by the widening variance of the graph of Panel A of our non-patrilineal (NPT) simulations.

II. Reviewer #2: Comments to the Authors

This manuscript examines the effect of kinship and social structures on human genetic diversity through mathematical models. Specifically, it attempts to find an explanation to the enigmatic male specific bottleneck dating to the last 5-7KYA that had been revealed by recent studies comparing Y chromosome and mtDNA diversity of human populations from across different continents. The key novelty of the work lies in the proposal of the model of competition between patrilineal corporate kinship groups to explain the sex-specific patterns in empirical data. This proposal relies on the synthesis of wide range of anthropological, archaeological and genetic evidence in light of the results of computer simulation results generated by the authors. The manuscript is well and clearly written, the topic should be of general interest to the wide readership of the journal.

Authors' reply: Thank you for your comments, and your positive appraisal of our proposal.

- I cannot identify any major flaws in the arguments while I think that the balance of the contextual and novel material presented in the main text should be shifted towards the latter.

Authors' reply: We agree. We have reduced the sections on the development of political complexity to a single sentence (line 489-492) that makes the point that evidence does exist from Madagascar and Polynesia that kinship groups are superseded as the relevant unit of intergroup competition after the transition to large-scale societies.

- At the moment only one of the main figures represents novel findings supporting the proposed model. While the contextual evidence is important I think it can be presented in much more condensed form.

Authors' reply: We agree. See our reply to the first point.

- As for the competing models that could explain the bottleneck I also found that more clarity should be given to the distinction of the proposed competition between patrilineal kinship groups and the patrilocality case, is the latter part of the proposed model or can the authors effectively rule out that patrilocality with patrilineality alone without the competition between the corporate patrilineal groups could not explain the findings.

Authors' reply: We agree. Our replies to the third referee's first and second points address this.

III. Reviewer #3: Comments to the Authors

In their paper CULTURAL HITCHHIKING: COMPETITION BETWEEN PATRILINEAL KIN GROUPS AND THE POST-NEOLITHIC Y-CHROMOSOME BOTTLENECK (TIAN CHEN ZENG; ALAN J.AW2;, AND MARCUS W. FELDMAN), authors present an interesting work with a mathematic development regarding patrilineal kin that could explain the discrepancy between Y chromosome and mt DNA diversity in Eurasia.

Authors' reply: Thank you for your detailed comments, which have been very helpful indeed in helping us sharpen our paper, and your positive appraisal of our proposal.

I have some major comments:

- Zeng et al propose a model based on competition between patrilineal kin groups to explain Y chromosome genetic patterns in human populations. While such proposition is interesting, other socio-cultural processes may reduce the Y chromosome diversity and the authors should refer

more extensively to previous literature on this subject. For example, Smouse (genetics 1981) showed that a dynamics of lineal fissions may reduce greatly the effective size of the population, and Heyer and colleagues studied patrilineal populations in Central Asia and showed that the Y chromosome genetic diversity in these patrilineal (and patrilocal) populations is greatly reduced in comparison to cognatic (and patrilocal) populations, suggesting a strong impact of the patrilineal dynamics on these sex-specific genetic patterns. In addition, Heyer et al showed that the transmission of reproductive success is greater in patrilineal than in cognatic populations which may contribute to the reduction of Y-chromosome diversity in these populations. They developed several population genetics statistics (see Chaix et al, 2004; Chaix et al 2007(current Biol) Marchi et al 2017; Heyer et al 2016 et al, (and ref in the papers) in relation with patrilinearity with a special focus on imbalance of coalescence tree.

In particular, transmission of reproductive success may explain the particular shape of the coalescent tree described in Zeng et al paper (see figure 5 “tree of Y chromosomes” and Heyer et al, TIG). In other words, a dynamics of lineal fissions with extinctions of kin groups (linked to demographic stochasticity) and/or the transmission of reproductive success may be sufficient to lead to an important Y chromosome diversity reduction.

Of course alternative models can explain this shape of coalescent tree and are worth testing which is the aim of this paper. But in the introduction these alternative processes that may shape Y chromosome diversity should be clearly stated.

Authors' reply: We state these alternate hypotheses under the heading “Hypotheses that cannot explain the bottleneck,” and have attempted to improve the clarity of this section.

We suspect that purely patrilineal dynamics in the absence of intergroup competition, specifically the effect on increased transmission of reproductive success along the paternal line, may have contributed to the bottleneck, but cannot account for it entirely. If we place this factor in the context of what we know of the historical record, we believe that social stratification and the development of large-scale societies should have increased the transmission of social and wealth inequality, and thus reproductive success, and this should be visible in the Y-chromosomal record in the form of bottleneck intensification. (For the effects of large-scale society and stratification in social status and wealth on reproductive inequality and its transmission, see references we have added in lines 121-158). However, the period associated with the rise of regional polities or social stratification and large-scale societies is instead associated with the end of the bottleneck, and the temporal coincidence between the development of political complexity and the lifting of the bottleneck is very good in each region of Afro-Eurasia. This would be expected if the bottleneck had primarily been generated by intergroup competition and lineage extinction events, which would be severely damped by the development of political units that transcend the individual lineage, but would not be expected if social inequality leading to transmission of reproductive success had been the primary mechanism that reduced Y-chromosomal diversity.

We detail this argument further in lines 121-158. This is, strictly speaking, an indirect theoretical argument in light of circumstantial evidence, but we nevertheless find it to be quite convincing.

Further, authors should also explain how to test between alternative models. On the general theory behind the model developed by Zeng et al:

- - One of the hypotheses behind the model is that there is competition/war between lineages. There is, to my knowledge (except for old school social anthropologists) no evidence that patrilineal populations are more prone to war than non patrilineal populations. Indeed base on anthropological theory the contrary is expected: patrilineality increases exogamous behaviors; exogamy is mainly used to build up alliances and therefore to reduce

war/competition (Levi-Strauss' theory). Further, do we have any archeological evidence of such an increase of war/death when this social system is thought to have started in Europe?

Authors' reply: Strictly speaking, our proposal does not require intergroup competition to have increased in intensity during the transition to either the Neolithic or a patrilineal regime of sociopolitical organisation. This assumption is unnecessary for our proposal. Rather, the main effect of the transition to patrilineality is the non-random distribution of the deaths that take place during intergroup competition. We emphasize this in line 194 of our paper: "If the primary unit of socio-political competition is the patrilineal corporate kin group, deaths from intergroup competition, whether in feuds, raids or open warfare, are not randomly distributed, but tend to cluster on the genealogical tree of males."

In our models, the same rates of male death in intergroup competition are used in both the patrilineal and non-patrilineal simulations, emphasizing that the contrasting effects on diversity are due only to social organisation and not to the intensity of warfare increasing under a patrilineal organisational regime.

- In their model, women are chosen at random (they constitute a "common resource"). Although I understand that a model is always a simplification of reality, random mating is clearly not the case in patrilineal populations. Conversely, there is a strong tendency to have non-random mating with matrilineal kin and 41% of unilineal populations prefer to marry with a first degree cousin.

Authors' reply: We agree with this criticism. The models that we use involve this assumption for tractability purposes. The effects of complex interactions between male and female genealogies on the simulations may be explored in future papers.

- "On the other hand, the development of political complexity in chiefdoms and states tends to reduce the prominence of corporate kin groups. They may not entirely have disappeared, but their salience as social identities and relevance as units of mobilization in political competition must have been reduced if sustained increases in social scale were to be achieved (Richerson and Christiansen (2013), also see chapter 2 of Yooe (2005)). Increased market integration under state peace also tended to catalyze the eventual breakup of descent groups in cultures with long histories of state rule (Korotayev, 2003)."
This is clearly an "old fashion" vision of social systems and too much in an "evolution" perspective with a direction in the evolution of social systems. Indeed, society can be unilineal with complex political systems.

Authors' reply: We agree that, in general, this historical trend that we highlight may be a very broad generalisation. However, the aspect relevant to our model of the development of large-scale societies is the production of relatively peaceful relations between unilineal social groups, even when they do exist. We do not deny that unilineal kinship structures may persist in large scale societies--indeed we mention this in lines 235-239:

"Once they emerge, complex societies, such as chiefdoms and states, tend to supervene the patrilineal kin group as the unit of intergroup competition; while they may not eradicate them altogether as sub-polity-level social identities, open conflict between such kin groups is suppressed very effectively (Otterbein and Otterbein, 1965; Otterbein, 1968)."

Instead, we emphasize that they may no longer be the most relevant units in intergroup competition, which is what produces the non-random pattern of male deaths when placed in genealogical trees.

More technical problem in relation with the model:

- Women (as a common resource) are incorporated in the “simple” Lotka-Volterra approach. However the population dynamics of women being solely considered as dependent on a growth rate and a carrying capacity that are independent of the dynamics of the other (male) populations, it does not seem to have any impact on the total dynamics. It seems that replacing the female dynamics by constant K would yield the same results, make the same point, and render the understanding easier to fathom.

Fair point. We agree that replacing female dynamics by a constant will yield the same qualitative results (i.e., decline in diversity). The reason we chose our model is to show that variations in female population size, which play a role in the dynamics of male populations, do not change the consequence of male competition. (Instead of stating this without proof, we'd rather show it.) Moreover, although not explicitly discussed, a more subtle mathematical distinction lies in the rate of convergence to the fixed points / steady state solutions: from the mathematical analysis, it is clear that this depends locally on the changes in the female population size.

- For non-specialists of the L-V competition equation, a **phase portrait** for the model with two or three male populations, would make it clearer to readers where the equilibria are located, whether they are stable or unstable and that some male populations will go extinct for any possible initial population.

Thank you for the fair point. We agree that having a phase portrait will help non-specialists understand the equilibria, their stability, and also the effect competition has on the dynamics. However, we have not included a phase portrait in our revision for the following reasons. First, the modified L-V system is only a qualitative model for our hypothesis, whereas our computational grid model much more realistically encompasses our hypothesis. (A schematic has been provided for our computational grid model; see Figure 4.). Second, in the interest of optimizing the use of space, we have tried very hard to include only information that is most directly relevant to justifying our hypothesis. Third, our phase portrait resembles the 2D phase portrait of a classical L-V system of two competing species, which can be found in many textbooks in ecology. Thus its inclusion in our revision is, in our opinion, less novel than an inclusion of a schematic that explicitly illustrates our computational model.

- Given that in the more “complex” computational grid model, the number of females is considered constant, why not use the same parameter for the simpler LV model ?
- It seems that having the parameter of the Poisson law of killed individuals indexed on $\sqrt{N_{c,h}(t)}$ has a strong effect on the dynamics. Why using root square? (Is this an algorithmically-driven or anthropologically-driven relationship). To which extent do the result depend on this parameter? Given this choice for the more “complex” computational grid model, why not use the same parameter for “simple” LV model (in which competition is related to $N_c(t)$ and not $\sqrt{N_c(t)}$).

Authors' reply: We used a square root law, because a linear law would result in groups of medium-to-large size declining at the fastest rate, and groups with small sizes declining at slower rates, with the smallest groups declining the slowest. This is because the rate of decline is proportional to the group size itself, and this proportionality plays a very strong role if the law is purely linear. This is very divergent from real-world conflict and is contrary to theoretical treatments of intergroup conflict in anthropology, the military sciences and cliodynamics, where group size is seen as strongly positively correlated with victory in engagements, group mobilization capacity produces great payoffs, and small group size places groups at a competitive disadvantage. We cite Turchin and Gavrilets (2009), Turchin (2009), and Kosse (1994) to justify our inclusion of such a size-dependent dynamic in our model.

We utilize an intuition that a group has to occupy some spatial area proportional to population size because of resource availability and distribution, but only competes at the edges of the

territory, such that the intensity of competition scales proportionally to the group perimeter but population scales proportionally to the group area occupied, which would mean that the rate of loss of males scales as the square root of the rate of growth of the population, i.e. population size.

- I may be mistaken, but it seems that another issue with indexing on the square root, is that the number of individuals killed in clan c, is

$$K_c(t) = \sum_h K_{c,h}(t) = Pol\{\alpha_c \cdot \rho_c \cdot [\sum_{c' \neq c} N_{c',h'}(t)] [\sum_h \sqrt{N_{c,h}(t)}]\}$$
 which contrary to the case where the indexation is on $N_{c,h}(t)$, is NOT the same r.v. as $Pol\{\alpha_c \cdot \rho_c \cdot [\sum_{c' \neq c} N_{c',h'}(t)] [\sqrt{\sum_h N_{c,h}(t)}]\}$. And so, to my mind, contrary to what is stated on page 5 of the SI, I am under the impression that “The mean proportion killed per generation of a group c is NOT proportional to the product of the total number of individuals not belonging to c and the square root of the number of individuals belonging to c ». Could you clarify this point ?

You are correct, we acknowledge an error on our part here. This has been corrected in the revision.

- Mutation: the process call “mutation” does not create here new mutation. This is a simple move of individuals from one haplogroup to the other. This is very intriguing.

Thank you for the positive comment.

- Mutation does not create new haplogroups but consists in the translation of individuals from one haplogroup to all others. If we understand correctly, whilst the set of all possible haplogroups/haplotypes is fixed, most of these sets are empty at first (300/400). What is the effect of the ratio of $|H|/|C|$ (which is 4-fold in your model) on the resulting dynamics?

You are correct. We attempt to approximate a scenario with potentially infinitely many haplogroups to mutate to, in concordance with models of mutation assuming infinite alleles or sites. When we modify the ratio of C:H (currently 1:5) we find that the results do not change qualitatively. We include figures demonstrating this in Subsection C.1 of the supplementary material (C19, C22, C23, C25) on pages 26-35.

- Mutation only affect the “major haplogroup”. Again, is this an algorithmically driven choice ? What would be the effect of allowing all mutations for each haplogroups ?

Yes. We think of mutations happening equally likely among all haplogroups, and consider the effect of such mutations on the aggregate level. We are assuming that no mutant is more likely to occur than any others, so that each mutation event from one haplogroup to any other is equally probable. This crucial assumption implies that roughly on average, the mutations cancel each other out, which is equivalent to a scenario where the major haplogroup undergoes mutation and mutants are distributed equally likely among the other haplogroups.

We have not tried allowing all mutations for each haplogroup, but this more accurate mechanism is likely to produce qualitatively similar dynamics based on the argument in the preceding paragraph. The cost of such an implementation is a significant increase in computer run time.

- Why do you assume that the fitness of a cultural group is linearly related to its group number? There is already advantage based on the size of the group, since death rate is proportionate to the square of the size of the group. (and again, why the square of the group size?). What are the dynamics in a model where all cultural groups have the same fitness, but with the square root effect favoring larger groups ?

Your comments are valid. The presence of a fitness differential, in the form of a linear law which amounts to the presence of selection amongst cultural groups, was indeed incorporated into our models. In the main text, we have now analysed a null model in which there is no longer a fitness differential, which keeps the model in line with our description of the sociocultural process. We find that while the spreading out of various haplogroup lineages is slowed down, they eventually die or persist in the same qualitative manner as they do in our original simulations. In other words, reductions in diversity were slowed at the beginning, but remained qualitatively similar by ~60 generations. We report these simulations in Section 3 of the supplement.

In addition to this null the model, we also address in Subsection 3.1 of the supplementary material the case where selection is present, showing that it does not affect the qualitative behaviour of the haplogroup diversity and dynamics. (Indeed this is just our previous model, which had led to such a conclusion.)

- Weeding: what is the logic to kill all groups that have less than 20 individuals? What is the importance of that threshold ?

We suggest that 20 is a reasonable estimate of a minimally viable cultural group.

- Replenishment: why half of the group is moved to a new grid cell ? The anthropological/ecological motivations for this portion of the algorithms are not clear

We do not wish to introduce directional dynamics in group size and group number into the simulation, which would occur if no fission took place and an increasing number of rows (cultural groups) were allowed to go empty (become extinct), while the remaining cultural groups become larger with time. We wish to assume a steady-state for group size and cultural group number (cultural diversity), which appears to us to be a good baseline assumption. Allowing directional change in group sizes and group number may result in a greater fit to real-world phenomena (as complex societies with millions of individuals from diverse kinship groups do in fact arise in the historical record) but this would add many complications to our model. Instead, we treat the situation of large-scale societies with diverse kinship groups as theoretically represented by a separate category of simulations, the nonpatrilineal simulations. We do this because the relevant unit of intergroup competition between large-scale societies is not the kinship group, and this is the aspect of theoretical relevance here.

Other major comments:

- The paper is too long, the part on Madagascar (although interesting) is out of scope of the paper. Idem for the part on Polynesia. It would need to be the object of another paper with genetic data.

Authors' reply: We agree. We have reduced this section to a single sentence (line 489-492) that emphasizes the point that evidence does exist from Madagascar and Polynesia that kinship groups are superseded as the relevant unit of intergroup competition after the transition to large-scale societies.

- Flatness of descent: other population genetics statistics have already been developed in regards with Y-chromosome genetic diversity in patrilineal population (Chaix et al Curr Biol 2007, Marchi et al, AJPA 2017). More work is needed to propose and validate a new population genetic statistics. I although guess that what you call flatness of descend is simply a ratio of coalescent branch length.
How it compares to other statistics that have already been developed to infer patrilineality?

Authors' reply: We agree with your comment, and have changed the language in section 5.3 (references to "flatness of descent" become "shallow coalescence") to reflect this. We are also

doing ongoing work to quantify “shallow coalescence” as well as what it would mean comparatively in human datasets, and the implications we can draw from it about social structure, in the same vein as Chaix and Marchi.

Minor comments on the models:

- In the presentation of the LV model (in the main text, p.7), the condition on the competition parameters for extinction of one population is expressed as “ $c_{12}, c_{21} > c_{11}, c_{22}$ ”. Is this equivalent to $\min(c_{12}, c_{21}) > \max(c_{11}, c_{22})$. If it is the case, the latter expression is easier to understand. However then, how does this mathematical condition differ from the literal condition “if interpopulation competition dominates intrapopulation competition” expressed a few lines later. It seems to be equivalent, and this should (if it is the case) be made clearer.

We agree and have made the changes. Thank you for the suggestion.

Note that while the mathematical condition is the same as “if interpopulation competition dominates intrapopulation competition,” the competitive exclusion principle applies only to standard competitive Lotka-Volterra systems and not our model in particular. We add a preceding clause to that sentence (line 285-286) to reflect this.

- The reference to the “competitive exclusion principle” of Hofbauer and Sigmund would benefit from further explanation on the role of women as “resources” and why the other populations (males populations) do not constitute “resources”. I suppose it is related to the domain of definition of parameters c_{ij} , but this would make it easier to understand were that point to be clarified.

Yes, indeed it has to do with the domain of definition. We have included a note for the reader to consult Hofbauer, where the domain of definition is explicitly described. Thank you.

- In the algorithm description of the “computational grid model”: Is it correct that in step 3.ii it is N_{c,h_c} that is reduced by the mutation? Shouldn't it be $N_{c,h_c}^{unscaled}$?

Yes, we have made a correction. Thank you for pointing this error out.

- In step 6 (replenishment), it is not clear what happens in the case of several groups c having $\sum_h N_{c,h}(t+1) = 0$.

We pick the largest cultural group and split it to fill in empty rows. Figure 4 provides a depiction of all steps in our model.

- Figure 2,3,4,5 supp material. One point is not clear: figure A show that some haplogroups disappear (for example in fig2, in some case the value is lower than 100) therefore I would expect in figure on the right to see some haplogroups reaching a value of zero.

Yes, this is a pointed observation. Our graphs show the results of the *averaging* of 100 simulations, where in each simulation a different lineage hits zero. Thus, while the haplogroup richness declines on average, the mean trajectories of each haplogroup over 100 simulations do not hit zero.

To ensure that the reader is not confused, we have now included in our graphs (1) haplogroup trajectories for a single simulation of the process, and (2) the average haplogroup trajectories over 100 simulations.

15 March 2018

- Correct haplotypes in figure/haplogroups in legend (figure 3)

Thank you for pointing this out. They have been corrected.

REVIEWERS' COMMENTS:

Reviewer #1 (Remarks to the Author):

After reading the revised paper it is clear to me that the authors dutifully and, I think, successfully addressed the concerns I had raised during the first round of reviews. I still find the paper of interest, and I think it might stimulate new and interesting research into the effects of intergroup competition in structured populations. The additional simulations reported in the revised version provide a better understanding of the processes of interest, though I think the authors would agree that this is just the beginning of what could be a fruitful trajectory of theoretical work. Indeed, I find it likely that this paper will help spark future work in this direction. Whether or not the conclusions presented here find more or less support as a result of future study remains to be seen, of course. However, I think the ideas are supported well enough by the proof of concept models presented here to be published and exposed to the scientific community for further scrutiny. The revised manuscript benefits from cutting a large section of tangentially related text. Although one might still find a couple of other places that could be shortened by a paragraph or two, overall this version is leaner, more focussed, and thus more accessible than the original.

Reviewer #3 (Remarks to the Author):

You have answer all my comments

24 April 2018

Author's Replies and Revisions

I. Reviewer #1: Comments to the Authors

After reading the revised paper it is clear to me that the authors dutifully and, I think, successfully addressed the concerns I had raised during the first round of reviews. I still find the paper of interest, and I think it might stimulate new and interesting research into the effects of intergroup competition in structured populations. The additional simulations reported in the revised version provide a better understanding of the processes of interest, though I think the authors would agree that this is just the beginning of what could be a fruitful trajectory of theoretical work. Indeed, I find it likely that this paper will help spark future work in this direction. Whether or not the conclusions presented here find more or less support as a result of future study remains to be seen, of course. However, I think the ideas are supported well enough by the proof of concept models presented here to be published and exposed to the scientific community for further scrutiny. The revised manuscript benefits from cutting a large section of tangentially related text. Although one might still find a couple of other places that could be shortened by a paragraph or two, overall this version is leaner, more focused, and thus more accessible than the original.

Authors' reply: Thank you for your positive comments. We agree that our work opens doors to both theoretical work and future studies that amalgamate more genetic data with anthropological theory, archaeological and archaeogenetic findings, and quantitative models. We are also happy to have our models subject to scrutiny.

II. Reviewer #3: Comments to the Authors

You have answer all my comments.

Authors' reply: Thank you, we are pleased to hear that.